



# A Climate Data Record (CDR) for the global terrestrial water budget: 1984-2010

Yu Zhang[1], Ming Pan[1], Justin Sheffield[1], Amanda Siemann[1], Colby Fisher[1], Miaoling Liang[2], Hylke Beck[1], Niko Wanders[1], Rosalyn MacCracken[3], Paul R. Houser[3], Tian Zhou[4], Dennis P. Lettenmaier[5], 5 Yingtao Ma[6], Rachel T. Pinker[6], Janice Bytheway[7], Christian D. Kummerow[7], Eric F. Wood[1]

[1]Department of Civil and Environmental Engineering, Princeton University, Princeton, NJ 08544, USA
[2]National Meteorological Center, China Meteorological Administration, Beijing, 100081, China
[3]George Mason University, Fairfax, VA 22030, USA
[4]Pacific Northwest National Laboratory, Richland, WA 99352, USA
[5]Department of Geography, University of California - Los Angeles, Los Angeles, CA 90095, USA
[6]Department of Meteorology, University of Maryland, College Park, MD 20742, USA
[7]Department of Atmospheric Science, Colorado State University, Fort Collins, CO 80523, USA

*Correspondence to*: Ming Pan (mpan@princeton.edu)

**Abstract.** Closing the terrestrial water budget is necessary to providing consistent estimates of budget components for
understanding water resources and changes over time. Given the lack of in-situ observations of budget components at anything but local scale, merging information from multiple data sources (e.g. in-situ observation, satellite remote sensing, land surface model and reanalysis) through data assimilation techniques that optimize the estimation of fluxes is a promising approach. In this study, a systematic method is developed to optimally combine multiple available data sources for precipitation ($P$), evapotranspiration ($ET$), runoff ($R$) and the total water storage change ($TWSC$) at 0.5° spatial resolution globally and to obtain
water budget closure (i.e. to enforce $P - ET - R - TWSC = 0$) through a Constrained Kalman Filter (CKF) data assimilation technique. The resulting long-term (1984-2010), monthly, 0.5° resolution global terrestrial water cycle Climate Data Record (CDR) dataset is developed under the auspices of the National Aeronautics and Space Administration (NASA) Earth System Data Records (ESDRs) program. This dataset serves to bridge the gap between sparsely gauged regions and the regions with sufficient in-situ observations in investigating the temporal and spatial variability in the terrestrial hydrology at
multiple scales. The CDR created in this study is validated against in-situ measurements like river discharge from the Global Runoff Data Centre (GRDC) and the United States Geological Survey (USGS) and *ET* from FLUXNET. The dataset is shown to be reliable and can serve the scientific community in understanding historical climate variability in water cycle fluxes and stores, benchmarking the current climate, and validating models.

## 1 Introduction

Quantification of the terrestrial water budget and its evolution over time at fine spatial resolutions is critical to understanding the availability and variability of Earth's terrestrial water budget and the exchanges and interactions among the terrestrial,




atmospheric and oceanic branches of the hydrosphere, and to assess the risk of hydrological extremes such as floods and droughts at regional to global scales. Understanding the mean state and variability of the terrestrial water budget is also one of the primary goals of World Climate Research Programme's (WCRP) Global Energy and Water EXchanges (GEWEX, (Morel, 2001)) Project and the National Aeronautics and Space Administration (NASA) Energy and Water cycle Study (NEWS:

(NASA NEWS Science Integration Team, 2007). The overarching goal of GEWEX is to "reproduce and predict, by means of suitable models, the variations of the global hydrological regime, its impact on atmospheric and surface dynamics, and variations in regional hydrological processes and water resources and their response to changes in the environment, such as the increase in greenhouse gases" (http://www.gewex.org/gewex_overview.html). The grand challenge of the NEWS project is "to document and enable improved, observationally-based, predictions of energy and water cycle consequences of Earth

system variability and change" (http://www.nasa-news.org). Toward these goals, a number of Earth System Data Records (ESDRs) for the major components of the terrestrial water budget are developed under NASA's Making Earth Science data record for Use in Research Environments (MEaSUREs) program. While the MEaSUREs program refers to long-term, satellite based data records as Earth Science Data Records (ESDR), they are generally referred to as Climate Data Records (CDR) following the National Research Council report where a CDR is defined as "a time series of measurements of sufficient length,

consistency and continuity to determine climate variability and change" (National Research Council, 2004). We will refer to the data set developed and described in this paper as a CDR.

The terrestrial water budget consists of four major components: precipitation ($P$), evapotranspiration ($ET$), runoff ($R$) and total water storage change ($TWSC$) as shown in Eq. (1). The total water storage change over a time interval ($TWSC$) is balanced by the difference between the incoming water flux of precipitation ($P$) and outgoing water fluxes of evapotranspiration ($ET$) and

surface and sub-surface runoff ($R$) for a control volume from the Earth's surface to a lower bound at depth:

$$TWSC = P - ET - R \qquad\qquad (1)$$

In-situ observations are often considered as the ground "truth" to quantitatively estimate the water budget terms. However, limited network coverage, especially for data sparse regions, has resulted in a long-time challenge for assessing the terrestrial water budget. Presently, satellite remote sensing has become a major data source to measure the various terms because of its

generally global coverage and sufficient temporal repeat times. A number of satellite based products have been developed to estimate precipitation over the globe, including the Tropical Rainfall Measuring Mission (TRMM) Multi-satellite Precipitation Analysis (TMPA: (Huffman et al., 2007;Huffman et al., 2010)), the Precipitation Estimation from Remotely Sensed Information using Artificial Neural Networks-Cloud Classification System (PERSIANN-CCS: (Hong et al., 2007)), and the Climate Prediction Center morphing method (CMORPH: (Joyce et al., 2004)). For evapotranspiration, global estimates can be

derived from a combination of satellite surface radiation budget (SRB), surface meteorology and vegetation cover (Fisher et al., 2008;Mu et al., 2007;Vinukollu et al., 2011;Zhang et al., 2010;Zhang et al., 2015). With the NASA Gravity Recovery And Climate Experiment (GRACE) mission, launched in March, 2002, (Landerer and Swenson, 2012;Tapley et al., 2004;Wahr et al., 2004), the changes in gravity detected by the GRACE satellites can be used to derive estimates of TWSC, albeit at relatively coarse scale. GRACE has been widely used to study changes in the terrestrial water storage (Rodell et al., 2009;Rodell et al.,





2011); the terrestrial water budget (Long et al., 2014a;Long et al., 2014b;Pan et al., 2012;Sahoo et al., 2011;Gao et al., 2010;Sheffield et al., 2009;Wang et al., 2014) and hydrological extremes such as droughts (Thomas et al., 2014;Famiglietti, 2014). For runoff, earlier studies estimated the global mean terrestrial runoff by simply calculating the differences between precipitation and evapotranspiration under the assumption of negligible long-term total water storage change (Berner and

5 Berner, 1987;Baumgartner and Reichel, 1975). But this "inferred" runoff estimation approach can only be applied to estimate the long term mean since water storage change cannot be neglected at short temporal scales, e.g. daily, monthly or seasonally. Furthermore, human interaction with the storage might also play an important role. For example, reservoir filling after construction, inter-annual reservoir storage changes, and groundwater pumping (Rodell et al., 2009;Famiglietti, 2014;Voss et al., 2013) can significantly contribute to observed storage changes at regional scales. As an alternative, river discharge can be

estimated from satellite altimetry (Birkett et al., 2002;Berry et al., 2011), for example, the future Surface Water Ocean Topography (SWOT) (Durand et al., 2010) mission. These satellite missions provide a promising and cost-efficient way of estimating individual water budget components. However, when combined together, they do not close the water budget because of errors in the individual component estimates. (Sheffield et al., 2009) found that high bias in satellite precipitation, particularly in the summer, was the major factor in budget non-closure over the Mississippi River basin. (Gao et al., 2010) also

concluded that water budget closure over 13 large continental rivers in the US was not achieved using remote sensing data mainly due to the biases in precipitation and *ET*.

In addition to space-borne satellites, our understanding of the hydrological cycle in data-scarce regions has also depended on other data sources such as land surface models (LSM) (Trenberth et al., 2007;Trenberth and Fasullo, 2013b), and weather/climate reanalysis (Reichle et al., 2011). Off-line LSM simulations can provide long-term budget estimates with

20 closure by design (Nijssen et al., 2014;Sheffield and Wood, 2007;Trenberth et al., 2007;Oki and Kanae, 2006). Reanalysis model output provides information that can be used to estimate the water budget at basin to continental (Betts et al., 2003a;Betts et al., 2003b;Betts et al., 2005) and global (Reichle et al., 2011;Balsamo et al., 2015) scales. These large scale land surface and reanalysis models have pushed the global water budget inventories into a new era where sparse traditional in-situ observations are supplemented.

However, different types of uncertainties exist in these sources of information including those in the parameterizations (satellite retrieval algorithms, LSM and reanalysis process representations), in LSM parameters such as soil and vegetation properties, and forcing data (surface radiation and meteorology) and in reanalysis data assimilation procedures. Therefore, an optimal "combination" of all data sources, including in-situ and remote sensing, LSM and reanalysis data, with their extensive spatial coverage and fine resolution, has the potential to overcome the limitation of relying on a single data source, and to offer

improved accuracy, spatial and temporal coverage, and consistency in creating long-term, large scale water budget information at fine spatial resolutions (Pan et al., 2012).

To address the non-closure problem, techniques have been developed to assess the uncertainties of each budget component and to enforce water budget closure from either multiple data sources (Pan et al., 2012) or single source (Sahoo et al., 2011), usually at the scale of major river basins across the globe. For example, (Rodell et al., 2015) recently quantified the mean





annual and monthly water budgets over continents and ocean basins for the first decade of the 21$^{st}$ century by using data sets that combine satellite remote sensing and conventional observations.

Building on an increasingly available inventory of global water budget data sets from in-situ, satellite, reanalysis and land surface models, this study reported here has five advances over previously reported work. These are to: (1) expand the use of

the Constrained Kalman Filter (Pan and Wood, 2006) data assimilation technique in closing the water budget from that reported by (Pan et al., 2012) and (Sahoo et al., 2011), (2) extend the data records back in time to 1984 (versus to 2000 in (Rodell et al., 2015) and forward to 2010 (vs previous analyses which usually stop near the turn of the 21$^{st}$ century), (3) refine the spatial resolution to 0.5° for the land surface (versus basin scale analysis in (Pan et al., 2012) and (Sahoo et al., 2011), and continental and oceanic analysis in (Rodell et al., 2015) and (Trenberth and Fasullo, 2013a)), and account for the oblateness of Earth, (4)

develop a harmonized global terrestrial water cycle CDR by merging the full combination of in-situ and satellite remote sensing observations, LSM simulations, and reanalysis model outputs at monthly and 0.5° spatial resolution for the period 1984 – 2010. The CDR data set includes estimates for all major terrestrial water budget variables (i.e. *P*, *ET*, *R* and *TWSC*) with budget closure at the grid-scale, and finally (5) validate the CDR against in-situ observations not used in the development of the data set.

To the authors' knowledge, this paper presents the first attempt to estimate over multiple decades the global terrestrial water budget (Greenland and Antarctica excluded), with closure at a 0.5° grid scale using this diversity of observational data sources. The data set provides comprehensive and detailed information for water budget analyses over land, and will be of particular significance in those sparsely or ungauged regions for understanding historical climate variability of the water cycle, and for benchmarking and validating climate models.

In developing the dataset, significant challenges were faced that need to be addressed. These included: (1) How consistent are the different products at different spatial scales? (2) What is the best approach to assess the uncertainty of each individual product and then optimally merge them? And, (3) what is the spatial and temporal variability of the non-closure errors and how to attribute them? Given the developed CDR, a key question is whether the merged dataset is in agreement with in-situ observations, and thus be able to capture historical hydro-climatological events (e.g. floods and droughts)?

Section 2 introduces the data sources and the methodology. Section 3 carries out a consistency and uncertainty analysis for the multiple input data sources, and investigates the spatial variability of the non-closure errors and their attribution during the budget closure enforcement process. Budget estimates based on the closure constrained dataset are presented at global, continental and large basin scales. Then the CDR is validated against in-situ runoff and *ET* in section 4. Conclusions from the research and future work are discussed in Section 5.

**2    Data Description, Analysis and Methodology**

In this study, the water budget is estimated and constrained at 0.5°, monthly, for the global land area excluding Antarctica and Greenland. In addition, continental and basin-scale budget estimates are also provided, including six continents and 32 major





basins (Figure 1) from across the world with a range of climatic regimes. Information about the input data sources (data length, original spatial and temporal resolution, and references) is listed in Table. 1. The Community Land Model (CLM) and NOAH land surface models are used for seasonal cycle analysis but are not included later in the merging and constraining algorithm because of significant disagreement between their seasonal cycles and observations, as discussed in sections 2.1.3 and 2.1.4.

The 27-year period is divided into four consecutive sub-periods (1984-1997, 1998-2002, 2003-2007, and 2008-2010) based on the data availability and overlap (Table. 2). Note that the total water storage from GRACE for the initial year 2002 is excluded from the study due to missing values.

## 2.1 Input Datasets

### 2.1.1 Precipitation

A set of precipitation products is evaluated including the remote sensing precipitation product from Colorado State University (CSU: (Bytheway and Kummerow, 2013)) with uncertainty estimates, the gauge-based Global Precipitation Climate Centre (GPCC) product (Schneider et al., 2014), the multi-source merged products of the Princeton Global Forcing dataset (PGF: (Sheffield et al., 2006)), and the Climate Hazard group InfraRed Precipitation with Stations (CHIRPS: (Funk et al., 2014)). Please refer to Supplement I for more information on these data sets.

Figures 2 and 3 show the seasonal cycles of these four precipitation products over six continents and over twelve selected representative basins distributed in different continents and climate regimes, for their overlapping period of 1998-2010. The coefficient of variation (CV), calculated as the standard deviation divided by the ensemble mean, is plotted to quantify the uncertainties among the precipitation product ensemble of PGF, CSU, GPCC and CHIRPS. The CV is first calculated for each grid cell and then averaged over continents or basins. There is no spatial coverage beyond 50°N to 50°S from CSU or CHIRPS.

Therefore, only the grids between 50°N and 50°S are used to calculate the seasonal cycles in Figure 2. Likewise in Figure 3, only PGF and GPCC are compared over those basins which are either outside or extend poleward of 50°N (or 50°S) (e.g. Lena and Mackenzie river basins). Similar to the conclusion of (Pan et al., 2012), who examined a different set of datasets, the spread among these four products is higher in the densely gauged continents in Europe and North America (and basins in those two continents such as the Danube and Mississippi), with a CV ranging from 5-12% and 2-8%, respectively (Figures 2 and 3),

than in the sparsely gauged regions, such as Amazon (Figure 3). There is an "abnormal" high spread (high CV) for the Niger River basin (sparsely gauged) during the dry season because the ensemble mean of the four precipitation products, is close to zero (Figure 3). The uncertainties are also high for the Mekong River basin where the rainfall totals are high and dominated by the monsoon season (Figure 3). The high uncertainties in less densely gauged regions could originate from the different gauge densities from different products, or the ways in which the data are merged and gridded. It is interesting to note that the

average discrepancy between the highest estimates (CSU) and the lowest (CHIRPS) over Europe is around 15mm/month throughout the year (Figure 2). This discrepancy is more prominent at basin scales; for example, the monthly mean difference between CSU and CHIRPS in the densely gauged basins such as Danube and Mississippi is around 20 mm/month (Figure 3).



CHIRPS is a blended precipitation product (e.g. precipitation climatology, remote sensing from multiple sources, seasonal forecast form Climate Forecast System Version 2 (CFSv2), and in situ observations) but it is dominated by gauge corrections in regions with higher gauge density such as Europe and North America, and therefore in basins such as the Danube and Mississippi. The differences among the three gauge-merged products PGF, GPCC and CHIRPS might be possibly from the different data sources that they merge other than gauge observations, different numbers of gauges used and under-catch corrections. The seasonal cycles in Figures 2 and 3 are consistent with the climate regimes, e.g., the inversed seasonality in the Murray-Darling basin, the high peak in South America in March, and wet summer in low latitudes.

### 2.1.2    Evapotranspiration

Unlike precipitation with relatively dense in-situ observations, especially for developed regions, in-situ based evapotranspiration estimations (from flux towers) are very sparse. Here we collect ten gridded global terrestrial evapotranspiration ($ET$) products, of which five are satellite derived, two are reanalysis products and three are from land surface models. One satellite product is the Global Land-surface Evaporation: the Amsterdam Methodology (GLEAM: (Miralles et al., 2011). The other four satellite products are derived from two algorithms, the Penman-Monteith (PM) and Priestly-Taylor (PT), cross-combined with two forcing inputs, the SRB-CFSR (Surface Radiation Budget – Climate Forecast System Reanalysis) and SRB-PGF. These four products are referred as: SRB-CFSR-PM, SRB-CFSR-PT, SRB-PGF-PM and SRB-CFSR-PT (Vinukollu et al., 2011), the SRB-PGF forced Penman-Monteith (PM), and Priestly-Taylor (PT) $ET$ referred to here as SRB-PGF-PM, and SRB-PGF-PT (Vinukollu et al., 2011). The two reanalysis $ET$ products are from the ERA-interim (Simmons et al., 2006) and NASA's Modern-Era Retrospective Analysis for Research and Application (MERRA: (Rienecker et al., 2011). The LSM $ET$ datasets are from the Variable Infiltration Capacity model (VIC v4.0.6), CLM v3.5 and NOAH v3.4 forced by an updated version of PGF. Please refer to Supplement I for more information.

The ten $ET$ products show less consistency in the seasonal cycle (Figure 4 and 5) than the precipitation datasets (Figures 2 and 3). At continental scales (Figure 4), the reanalysis $ET$ products (ERA-Interim and MERRA) generally have relatively high values for the six continents, while the LSMs generally predict lower values over Asia, Europe and North America. Most of the satellite $ET$ products (i.e., GLEAM, SRB-CFSR-PM, SRB-CFSR-PT, SRB-PGF-PM, and SRB-PGF-PT) lie between the reanalysis and LSMs in Asia, Europe and North America. More striking is the relative lack of consistency among those ten $ET$ products for the wet tropical basins (Amazon, Congo and Mekong). The seasonality of $ET$ over these basins is complex because of the overall energy limitation but seasonally and spatially varying moisture limitation (Guan et al., 2015). These results imply that the ten approaches have significant differences in their derived surface radiation budget and meteorology as well as the parameterizations of evaporative processes (potential $ET$, transpiration, interception, and soil evaporation) and their interaction with phenological and environmental controls. The relatively higher consistency of the remotely sensed algorithms for these basins is in part a result of using the same (or closely similar) surface radiation budget but different meteorological forcings.



### 2.1.3 Runoff

The three LSMs are forced by the same meteorological forcing from PGF to simulate global runoff over land. The VIC simulation was calibrated over 43 well-distributed major global basins against the measured streamflow data from the Global Runoff Data Center (GRDC, http://grdc.bagf.de) while CLM and Noah are un-calibrated. Please refer to Supplement I for

additional model information under the evapotranspiration section. Figures 6 and 7 display the seasonal cycles over the six continents and the twelve representative major river basins. Noah shows an obvious discrepancy in Europe and North America, which include high latitude regions, relative to VIC and CLM; while CLM shows large disagreement against VIC and Noah in Oceania (Figure 6). The disagreement between the LSMs can also be found at basin scales (e.g. Danube, Lena. Mackenzie, Yukon and Murray-Darling in Figure 7). Additionally, Figure 13 shows the verification of the runoff from the LSMs against

GRDC observations for 26 basins that have available data records longer than 3 years during 1984-2010. Noah shows a negative runoff bias against GRDC for most of the mid to high latitude basins (Columbia, Danube, Indigirka, Kolyma, Kena, Mackenzie, Northern Dvina, Ob, Olenek, Pechora, Yenisei and Yukon; Figure 13). CLM has better performance over high latitude basins than Noah but it shows a high overestimation of runoff for the Danube, and Don (Figure 13). None of the LSMs captures the seasonal cycles for the Indus and Senegal basins. Nonetheless, the authors recognize that runoff estimates using

a number of LSMs (e.g. (Haddeland et al., 2011)) can provide uncertainty estimates in simulated runoff. However, CLM and Noah runoff estimates are not merged into the CDR developed in this study in order to avoid the large biases from their un-calibrated parameters. Additional reasons for not merging CLM and Noah are discussed in section 2.1.4.

### 2.1.4 Total Water Storage Change

The total water storage change ($TWSC$) is taken from the LSMs and the GRACE data. The monthly total water storage anomaly

($TWSA$) time series from ReLease 05 (RL05) that are processed by three centers, Geoforschungs Zentrum Potsdam (GFZ), Center for Space Research at University of Texas, Austin (CSR), and Jet Propulsion Laboratory (JPL), are used to calculate the $TWSC$ via the backward difference equation in Eq.(2) and central difference equation in Eq.(3). Comparisons indicate that the central difference calculation Eq. (3) is in better agreement with VIC inferred $TWSC$. Therefore, the central difference $TWSC$ has been used.

$$TWSC = (TWSA_t - TWSA_{t-1})/\Delta t \qquad (2)$$
$$TWSC = (TWSA_{t+1} - TWSA_{t-1})/2\Delta t \qquad (3)$$

Even though VIC only computes the water storage in the upper few meters of the soil column (depending on the calibrated storage capacity in its $2^{nd}$ and $3^{rd}$ layers), this is the most active part of the soil column. Therefore studies (e.g. (Gao et al., 2010;Tang et al., 2010) found reasonable agreement between changes in $TWSC$ from GRACE and the VIC model.  Similar

results were also found in this study: $TWSC$ from VIC and GRACE (from GFZ, CSR, and JPL) are in good agreement at both continental (Figure 8) and basin scales (Figure 9) except for some timing lags in the high latitude basins of the Lena and Yukon. This lag between GRACE derived minimum $TWSC$ and VIC inferred minimum $TWSC$ suggests that the snowmelt





(and subsequent runoff) starts earlier in VIC than observed by GRACE, or that more snowmelt ponds into wetlands or discharges into lakes, neither of which are well represented in VIC as its snow melt discharges more directly into rivers. In contrast, Noah shows a reversed seasonal cycle in those high latitude continental regions and basins such as Asia, North America, Danube, Lena, Mackenzie and Yukon, while CLM shows disagreement in the seasonal cycle in Oceania as well as in the Danube and Mississippi basins relative to GRACE observations. Not surprisingly, the spread within the three GRACE products is very small compared to the differences against VIC (Figures 8 and 9). (Sakumura et al., 2014) found that the ensemble mean (simple arithmetic mean of JPL, CSR, GFZ) was the most effective method in reducing the noise in the gravity field solutions within the available scatter of the solutions. Therefore, the ensemble mean of the $TWSC$ from GFZ, CSR, and JPL is taken as the best $TWSC$ product derived from GRACE, and this is used in the later water budget analysis.

## 2.2 Methods

All the datasets, as listed in Table 1, are first aggregated or disaggregated to 0.5°spatially and monthly values using bilinear interpolation, then the errors/uncertainties of each product are assessed. Estimates for the same water budget variable are then merged following the algorithm described in (Luo et al., 2007). The merged water budget estimates are further adjusted to ensure closure at every grid using the Constrained Kalman Filter (CKF) approach of (Pan et al., 2012). Then, the unconstrained and constrained water budgets are analyzed at different scales. Figure 10 provides a flow chart of the procedure.

### 2.2.1 Uncertainty estimation and data merging technique

There is no best estimate or observation of each individual water budget component at the grid scale over the globe due to the limited spatial coverage of in-situ measurements. This is especially true for the evapotranspiration observations from the flux tower networks. Thus, the limited availability of gridded ground observations makes it impossible to quantify the error in each water budget component. Therefore, in this study, the deviation from the ensemble mean of all data sources for the same budget variable is used as a proxy of the uncertainty/error in individual products. The merging procedure for each budget component is a weighted averaging where the optimal merging weight $w_i$ is given by the following equation (Luo et al., 2007;Sahoo et al., 2011):

$$w_i = \frac{1}{\sigma_i^2} \Big/ \sum_{i=1}^{n} \frac{1}{\sigma_i^2} \tag{4}$$

in which $w_i$ is the merging weight for product $i$, $\sigma_i^2$ is the error variance of product $i$ calculated against the ensemble mean, and $n$ is the total number of products. Note that $\sum w_i$ equals to 1. The larger the error variance of product $i$, the lower is its weight.

The number of products merged into single water budget estimate varies in the different sub-periods due to the data availability (Table 2). A "data consistency adjustment" is applied after the data merging process in order to guarantee the consistency of the climate data record (CDR) estimated in this study. Taking precipitation as an example, first, for the period with complete data records (i.e. 1998-2008), the inner-annual monthly mean precipitation merged from all the available products (i.e. PGF,



GPCC, CHIRPS, CSU), and the mean precipitation merged from the available products (i.e. PGF, GPCC, CHIRPS) during the incomplete data records period (i.e. 1984-1997 during which the CSU is not available) are calculated, respectively. Then the inter-annual monthly climatological bias, which is the monthly mean precipitation merged from PGF, GPCC, CHIRPS and CSU minus that merged from PGF, GPCC and CHIRPS, is simply added to the inter-annual monthly mean precipitation during

the incomplete data records period (i.e. 1984-1997). The same procedure is then applied to adjust the data consistency for *ET* during 2008-2010 and *TWSC* during 1984-2002. We contend that this is a key step, as the temporal consistency of the CDR will impact the reproduction of historical hydrological extremes and the analysis of long-term trends for all the available water budget variables.

### 2.2.2    Enforcing water budget closure using CKF

In short, CKF redistributes the non-closure errors back onto the various water budget components according to their error levels and correlations. We define the water balance residual as *r = P – ET – R – TWSC*. If we write the budget components as a column vector *x*: $x = [P, ET, R, TWSC]^T$, then the residual of the water balance can be expressed as a linear function of the vector, *r = G x*, where *G* = [1, -1, -1, -1]. The error covariance matrix of *x* is calculated as $\varepsilon_{xx} = \overline{(\hat{x} - x)(\hat{x} - x)^T}$, where $\hat{x}$ is an estimate of *x*, its "true value". In this study, $(\hat{x} - x)$ is replaced with the spread of the ensemble in each water budget

component. This uncertainty estimation method was first proposed by (Adler et al., 2001) and then applied in (Tian and Peters-Lidard, 2010) to generate a global precipitation uncertainty map for a variety of satellite remote sensing products. $\varepsilon_{xx}$ has dimensions of 4 × 4 since *x* consists of 4 budget variables. Then the balance-constrained estimate is calculated from $\hat{x}' = \hat{x} - \varepsilon_{xx} G^T (G \varepsilon_{xx} G^T)^{-1} \hat{r}$. The residual term $\hat{r}$ is redistributed back onto the various water budget variables through the above equation. Mathematically, the CKF algorithm mimics assimilating a "perfect" (zero-error) observation of *r = 0*. Further details

are presented in (Pan and Wood, 2006).

## 3    Water Budget Merging and Constraint

### 3.1    Data Merging

All the products for the same water budget component are merged into a single estimate based on their uncertainties/errors relative to their ensemble mean as described in section 2.2.1. The values in Table 2 summarize the mean merging weights of

each individual product for different periods. Please refer to Figures S1 to S3 in Supplement II for the spatial maps of the merging weights from different products. The global mean merging weights for the precipitation are calculated over 50°N-50°S during 1984-1997 and 1998-2010. CHIRPS and CSU only cover 50°N-50°S; therefore, for those regions outside 50°N-50°S, PGF and GPCC are merged with equal weights (50%). Before the availability of the CSU product in 1998, the average merging weights of PGF, GPCC, and CHIRPS over 50°N-50°S (land) are 29.6%, 34.6% and 35.8%. CHIRPS is closest to the

ensemble mean especially for the Amazon basin, and therefore has a higher weight in that region (Figure S1 in Supplement





II). For the period 1998-2010 when CSU becomes available, CHIRPS (26.5%), GPCC (26.8%) and CSU (26.0%) have similar weights. Note that the weights vary with time and location. The annual mean of the merged precipitation is 767.0 mm for 1984-1997, 792.7 mm for 1998-2002, 786.7 mm for 2003-2007 and 802.9 mm for 2008-2010 (Table 3). Equivalent numbers at monthly scale are displayed in Figure 11 in terms of global maps. The values from Table 3 and Figure 11 are calculated
using the data consistency adjustment described in section 2.2.1.

For evapotranspiration, the averaged merging weights over land for each product are: VIC (11.3%), ERA (10.8%), MERRA (6.6%), GLEAM (12.8%), SRB-PGF-PM (17.2%), SRB-PGF-PT (15.9%), SRB-CFSR-PM (13.9%), and SRB-CFSR-PT (11.5%) during their common period 1984-2007 (Figure S2 in Supplement II). Among the eight *ET* products, MERRA has the lowest averaged merging weight as it has a relatively larger deviation from other *ET* products at both continental (Figure 4)
and basin scales (Figure 5). In the Amazon basin, MERRA shows nearly an opposite seasonal cycle against other *ET* products (Figure 5) and thus its merging weight is extremely low there (Figure S2 in Supplement II). The merged *ET* in the unconstrained budgets, averaged over land, are 518.0 mm/year, 523.6 mm/year, 516.0 mm/year and 522.0 mm/year throughout those four sub-periods (Table 3, the spatial maps can be found in Figure 11).

The runoff simulated from VIC is used as the "merged" terrestrial runoff at the grid scale since the gauge observations are
discrete and spatially incomplete. The annual averaged runoff over the globe is 338.9 mm/year during 1984-2010 (Table 3, see Figure 11 for the spatial maps for the sub-periods).

For the total water storage change, the uncertainty in VIC inferred storage change and GRACE derived storage change are simply assumed to be 5% and 10% of their actual values due to the lack of a better source for their validation (Pan et al., 2012). Consequently, the higher merging weight from VIC (67.1%) and lower merging weight from GRACE (32.9%) in Table 2 (and
Figure S3 in Supplement II for the spatial maps of merging weights) are a result of the assigned error ratios (i.e. 5% and 10%). Given the good agreement in *TWSC* between VIC and GRACE (Figures 8 and 9), the impact of such a subjective error assignment is relatively small. Globally, the monthly mean of *TWSC* is almost zero during the four sub-periods as shown in the fourth row of Figure 11. Nonetheless, multi-year variability due to drought and wet periods is observable. For example, the long-term drought in the central U.S. and Canadian prairies over the 1998-2002 period shows up as do the Brazilian
droughts in 1994-1995 and 2004-2005 that extended into Argentina (2004-2006). Also seen in Figure 11 is the wetting trend over the last two decades of the Sahel since the severe mid-1980's drought as well as the floods in Brazil in 2008.

### 3.2   Data Assimilation to close the water budget

The last row of Figure 11 shows the global maps of the non-closure errors for the sub-periods. The long-term, mean non-closure error relative to precipitation is around -9.8% over land during 1984-2010 (Table 3). The annual mean imbalance over
land ranges from -55.3 mm/year to -80.6 mm/year during the four sub-periods (Table 3).

Figure 12 shows an example of the unconstrained (left) and constrained (right) water budgets for the Amazon basin together with imbalances and their attribution (bottom). Over the Amazon basin where the total precipitation is large and the gauges are sparse, the precipitation uncertainty is higher. This results in precipitation being the main recipient of the non-closure error




attribution (third row on the right in Figure 12), receiving around 50% of the non-closure error for each of the sub-periods as well as the complete analysis period. Due to the "inconsistencies" in terms of different numbers of available data sources merged into the budget during the four consecutive sub-periods, the imbalance/non-closure error (third row on the left in Figure 12) does not show a regular seasonal cycle and a continuous pattern of imbalance.

The annual mean water budget in terms of $P$, $ET$, $R$ and $TWSC$ after water balance constraint are 781.8 mm, 463.9 mm, 318.0 mm and 0 mm during 1984-2010, respectively (Table 3). Note that direct application of CKF to enforce the water balance without other constraints may possibly lead to a non-zero $TWSC$ over long term, and sometimes a negative runoff. Therefore, two additional "filters" are added after the CKF. First, if the runoff is negative, we will re-run the CKF and only re-distribute the non-closure error onto the other three budget components. Second, for each grid cell if the long-term mean $TWSC$ over

1984-2010 is not zero, the monthly long term mean $TWSC$ will be subtracted from the $TWSC$ and added to the precipitation and evapotranspiration month by month during 1984-2010. Figure S4 in Supplement II shows the mean water budget components after the CKF water balance enforcement in addition to the mean water budget components before the enforcement in Figure 11. The long term mean of $TWSC$ at each grid cell is zero over the entire 27 years after the second filter, which is also named as "$TWSC$ de-trending". But the spatial variability of $TWSC$ still exists during the four sub-periods (Figure S4). A

more comprehensive comparison of the water budget estimation before and after the closure enforcement is listed in Table 4 at both the continental and basin scales. These water budget component values are spatially and temporally aggregated for each continent or basin over the analysis period of 1984-2010.

The attribution of the non-closure term for each water budget variable is based on the uncertainties among different products. The results from this study are in general agreement with (Pan et al., 2012) where the authors showed that $ET$ has a high non-

closure attribution in a large portion of the 32 river basins that they analyzed. The average attribution of non-closure errors to $ET$ over the globe is 45.4% during 1984-2010 compared to 38.4% for $P$, 4.9% for $R$ and 11.2% for $TWSC$ (see Table 3). For most of the regions $ET$ receives the highest attribution of the non-closure error, particularly, in Africa (50% attributed to $ET$ vs. 37% to precipitation, 3% to runoff, and 10% to $TWSC$; see Table 4), and Oceania (46% attributed to $ET$ vs. 41% to precipitation, 2% to runoff, and 10% to $TWSC$; see Table 4). Figure S5 additionally shows the global maps of the mean water

budget non-closure error attribution during different sub-periods. Higher attributions to precipitation occur in basins in mid to high latitudes such as the Danube (42% to precipitation vs. 38% to $ET$, 6% to runoff, and 12% to $TWSC$; see Table 4), and Don (42% to precipitation vs. 39% to $ET$, 3% to runoff, and 16% to $TWSC$; see Table 4), where the estimation of extreme rainfall rates remain less well resolved (Huffman et al., 2007;Yong et al., 2014). High non-closure attributions to precipitation also occur in tropical basins such as the Amazon (46% to precipitation vs. 33% to $ET$, 9% to runoff, and 12% to $TWSC$; see

Table 4) and Congo (46% to precipitation vs. 37% to $ET$, 6% to runoff, and 11% to $TWSC$; see Table 4) because the precipitation is large and the gauges are scarce in these basins. The attribution to the total water storage change is generally small except for the northern regions where snow, ice melt and seasonal storage changes in wetlands dominate the water budgets (Figure S5 in Supplement II). Runoff receives the smallest attribution of the imbalance among the four water budget





components for most regions over the globe, which is in agreement with what was concluded in (Sahoo et al., 2011). The mean attributions of each water budget component over different continents and basins over 1984-2010 are listed in Table 4 as well.

## 4 Validation of the MEaSUREs Global Terrestrial Water Budget CDR

The final CDR, which is the constrained global water budget with closure, is validated against in-situ observations in terms of

5 runoff and *ET* at multiple spatial scales.

### 4.1 Runoff Verification

In-situ river discharge observations are collected from three major data sources: (1) Global Runoff Data Center (GRDC), (2) United States Geological Survey (USGS), and (3) Australian Land and Water Resources Audit project (Peel et al., 2000). The observations were collected from GRDC for a total number of 32 large basins and 26 of them are used (as shown in Figure 13)

after filtering out those basins with less than three years of data during 1984-2010. Figure 1 provides the locations of these basins. 165 out of a total of 362 medium sized basins (5,000 to 10,000 km$^2$, 331 from GRDC and 31 from USGS) were selected for validations. For validation over small basins, discharge data for 862 basins (1,000 to 5,000 km$^2$) were collected from GRDC, USGS and the Australian Land and Water Resources Audit project. Basins under any one or more of the following conditions were excluded: (1) GRDC basins for which the catchment boundaries could not be reliably determined; (2) basins with large

dams (reservoir capacity greater than 10% of annual streamflow); (3) basins with urban areas greater than 2% (using the ''artificial areas'' class of the map from GlobCover, version 2.3; (Bontemps et al., 2011)); (4) basins with irrigated areas greater than 2% (using the Global Irrigated Area Map; http://www.iwmigiam.org); and (5) basins with either a gain or loss forest (change in land cover) > 20% of the basin area. For both the medium and small basins, those basins with data records length less than 5 years were also excluded. Figure 14(a) displays the locations of medium and small basins. The observed

discharge data were converted to runoff by dividing by the basin area upstream of the gauge location.

The seasonal cycles of runoff from the CDR created in this study over the 26 large basins, are compared against the GRDC observations as shown in Figure 13. Not surprisingly, the runoff estimated from the constrained system (grey dashed line) is not much different from the runoff estimated in the unconstrained system (which is VIC runoff shown by the solid blue line) as we assign a small error (10%) on the runoff component within the budget constraint algorithm. In general, VIC outperforms

the other two LSMs as VIC was calibrated over 43 major global river basins (Sheffield and Wood, 2007) although the calibration periods varied. Therefore, we believe that VIC can provide a reliable grid-scale estimate of runoff budget. Note that the seasonal peaks from Noah and VIC are in agreement for the Indus basin but their peaks precede the peak from the GRDC observations, which strangely happen in November. Comparing to other studies for the Indus River (Bookhagen and Burbank, 2010) show that the discharge peak occurs in the summer time , which is consistent with VIC and Noah. Likewise

for Senegal River, records from regional studies (Andersen et al., 2001) and (Stisen et al., 2008) show runoff peaks in August to September instead of April to May from the GRDC record. In summary, we believe that our CDR provides good runoff



estimates over the Amur, Danube, Mackenzie, Mekong, Mississippi, Pearl, Pechora, Yangtze and Yenisei rivers but unsatisfactory estimates over the Congo, Lena, Murray-Darling and Yellow rivers in that the predicted seasonal discharge differ significantly from the observed seasonal cycle. The reasons for this are due to water management not being included in the VIC model (e.g. Murray-Darling and Yellow rivers), a combination of scarce data and not including large wetlands (e.g. the Congo and Lena basins).

By filtering out those basins with non-significant correlations, Figure 14 compares CDR estimated runoff against in-situ observations for 165 medium basins and 862 small basins in terms of correlation coefficient (CC, Figure 14 (a) and (b)), and scatter plots (Figure 14 (c) and (d)), at the monthly scale. Again, the observed discharge measurements are converted to runoff using the basin area. 84 out of 165 medium basins (~ 51%) and 625 out of 862 small basins (~ 73%) have CC values that are larger than 0.5 as shown in Figure 14 (a) and (b). There are some medium basins with extreme low CC values (red dots in Figure 14(a)) in northern Canada where the lake/wetland influences are not modelled, and in south Africa where the sporadic rainfall is not picked up and the model fails to replicate the quick runoff. The MEaSUREs runoff from our CDR have CC values of 0.86 and 0.83 for the same medium and small basins as shown in Figure 14 (a) and (b), and have a Bias Ratio of 6% for medium basins and -16% for small basins (Figure 14 (c) and (d)). There is also a tendency for the model to underestimate runoff in the small basins in wetter regions (Figure 14 (d)). This scatter may be due forcing uncertainty, model calibration or omitted processes like water management (reservoirs, irrigation), all which might shift the timing of the runoff peak, particularly on a monthly basis. For the small basins, though they were filtered in an attempt to remove basins impacted by factors such as reservoirs, irrigation, urbanization, and so forth, they might be impacted by the scaling issues. The MEaSUREs CDR was computed at 0.5° grid resolution, which is approximately 50km near the equator. The small basins range from 1,000 − 5,000 km$^2$ so that the small basins only cover a maximum of two grid pixels and a minimum of 0.2 of a grid pixel for the smallest basin. The basin masks were extracted at a higher spatial resolution and then aggregated onto the 0.5° grids with the fractional area for the basin in order to minimize the impact of spatial mismatch. Nonetheless, the coarser spatial resolution of the CDR still affects the comparison of the runoff estimates with small-scale basin observations. No estimate of this resolution effect has been determined but the results shown in Figure 14(d) suggest that the effect is limited to a small number of basins.

## 4.2 *ET* Verification

Estimated *ET* is verified by two different approaches: first against an inferred *ET* that is computed from the difference between observed precipitation minus observed discharge (*P-R*) at the annual scale to minimize the effect of seasonal *TWSC*. This is done for the 25 large, 169 medium and 813 small basins which are selected by the criteria of no less than 5 years' annual records. And it is then secondly verified against in-situ observations from FluxNet tower data (Baldocchi et al., 2001).

The precipitation used in computing the inferred *ET* is from GPCC, which is a gridded rain gauge analysis that merges around 67,000 gauge measurements globally (Schneider et al., 2014). The observed runoff, *R,* is from the same sources as used in section 4.1. As shown in Figure 15, the correlation coefficients between MEaSUREs CDR *ET* and inferred *ET* from observed *P-R* are 0.97, 0.96 and 0.76 for those large, medium and small basins. For some of the MEaSUREs CDR *ET* over medium



basins, particularly wetter basins, do not match well with the observed *P-R,* are attributed to the effects of water management on our estimates of *R.* Essentially, if the CDR runoff that doesn't reflect water management is too large, then the estimates of *ET* will be too low, which is what is seen in Figure 15(b). The small basins show worse agreements with the inferred ET (with a Bias (%) of 20% for small basins versus 4% for large basins and -4% for medium basins, Figure 15) that we attribute to

scaling effects over estimating *R* than for the medium basins.

The *ET* estimates from the CDR are further assessed by comparing the grid-scale estimates with observations from 47 FluxNet towers, which measures the turbulent latent flux using the eddy covariance technique. Those 47 Flux Towers were selected based on data availability (Michel et al., 2015) in terms of the meteorological variables and radiations, and the final selection represents a variety of biomes and dry/wet climate regimes. The comparisons are made only over the summer (warm) seasons

for different time periods based on the data availability at each tower. The 47 flux towers are located in four continents (North America, Europe, Asia and Africa) as shown under different land covers that are defined by the International Geosphere-Biosphere International Program (IGBP, (Loveland et al., 2000)) in Figure 16(a). The tower stations are also described in Table 5. The validations against the FluxNet observations are only carried out during the warm season when *ET* is more dominant and when there are fewer missing values. From the 47 flux towers, we found out that our *ET* estimates from the CDR are in

high agreement with FluxNet observations under the land cover types WSA (Woody Savannas, one station in Africa and the other one in the US) and EBF (Evergreen Broadleaf Forest, only one station in France, Figure 16(b)). In general, our CDR *ET* matches well with the observation with a correlation coefficient of around 0.77 and a Bias Ratio of 11% except for some over estimations for the stations, most of which are under the land cover CRO (Cropland) and ENF (Evergreen Needleleaf Forest, Figure 16(b)). The positive Bias of MEaSUREs CDR *ET* relative to FluxNet observations are attributed to the tower

management – during the rainy days in the summer the flux towers are usually turned off thus underestimate the actual *ET* during the rainy days.

## 5    Conclusion and Future Work

A well-constrained, global inventory of the historical terrestrial water budget at fine resolution is essential to understanding the terrestrial hydrological cycle, its partitioning into individual components and their variability at regional to global scales.

In this study, the consistency and uncertainties of multiple hydrological data products are investigated, with precipitation found to have the highest consistency among the available products at both continental and basin scales compared to *ET* and *TWSC*. Data products from multiple sources that include in-situ and satellite remote sensing observations, land surface model estimates, and reanalysis model outputs are combined to create homogenized terrestrial water budget estimates at 0.5° spatial and monthly temporal scales for the period 1984-2010. This long-term water budget data record has both spatial and temporal

consistency, and is part of NASA's the Earth System Data Records (ESDRs) program. The CDR data set was created by applying a water balance closure constraint using the CKF data assimilation technique of (Pan and Wood, 2006). For the individual data products, their ensemble mean is taken as the best estimate for the variable, and the ensemble spread against





the ensemble mean as a proxy for their uncertainty. These estimates of the mean and uncertainty for the product are important assumptions underlying the development of data records. Nevertheless, we believe these data records represent the best, current knowledge for the global terrestrial water budget at the 0.5° and monthly scale over the 27-year period of 1984-2010. Additionally, the developed data set allows for the documentation of the water budget at continental and basin scales resulting

in a better depiction across multiple scales. The attribution analysis of the budget imbalance (non-closure) shows that $ET$ receives the largest adjustment in most regions –particularly in Africa and Oceania. In contrast, runoff tends to receive the lowest attribution of the non-closure error, in part due to the calibrated land surface model estimates from 43 large global basins. $TWSC$ receives larger adjustments in high latitude regions, which we attribute to the impacts from snowmelt and seasonal dynamics of wetlands and small lakes that are not well represented in VIC LSM.

The major challenge for the creation of ESDR/CDR of the terrestrial water budget (and potentially the terrestrial surface energy budget) is the lack of "ground truth" observations that can serve as reference data sets for bias correction. The sparseness of the observations in accessible data archives (e.g. GRDC: Global Runoff Data Centre for river discharge, GPCC: Global Precipitation Climatology Centre for precipitation and publically accessible and quality-controlled FluxNet data) is both a scientific and institutional challenge. Many additional gauge locations and data records exist and could contribute to the

development of improved CDR and our understanding of climate variability and change but have not been made available. Besides these operationally-focused observations, the relative inaccessibility of global FluxNet tower observations is also disappointing, although this situation has improved over the recent past. Even though there are over 650 towers in 30 regional networks covering 5 continents, the free fair-use subset of the La Thuile FluxNet dataset (which has been harmonized, standardized and gap filled) contains only 154 stations of which 47 were deemed useful for the validation presented here, based

on quality assessment (e.g. closure of the energy budget) and record length. Data availability and accessibility challenges need to be at the top of the agendas of the world's major space agencies (ESA, NASA, JAXA), international data programs such as the Global Climate Observing System (GCOS), Global Energy and Water EXchange (GEWEX) project of WCRP, Global Earth Observing System of Systems (GEOSS), and international agencies like the World Meteorological Organization. The "standard" statements and claims about "free and open access" to climate data from these programs have not resulted in

improved access. If the needed improvements to CDR are to occur, and must occur to better assess the impacts from global environmental change, improved in-situ data archiving and access by the scientific community is imperative for a more accurate analysis of climate variability and change.

The CDR developed in this study – the global terrestrial water budget at 0.5°, monthly for 1984-2010 is currently archived on our public server and available at

http://stream.princeton.edu:8080/opendap/MEaSUREs/WC_MULTISOURCES_WB_050/ and will be formally archived at the NASA Goddard Earth Science Data and Information Services Center (GES DISC) for the future use of climate and water management communities, and will advance our understanding climate variability and trends at multiple spatial scales.

As the authors are aware, essential directions in global water and energy cycle research are towards improved understanding historical climate, benchmarking future climate predictions, validating models, and improving the understanding of the





interactions among land, ocean and atmosphere hydrospheres. Future work will be targeted on extending the data sets to even longer period, and at finer resolutions, by combining upcoming new satellite missions and the analysis and predictions from more advanced modelling systems.

## 6    Acknowledgements

5    This study was made possible under the support of NASA grants NNX08AN40A (Developing Consistent Earth System Data Records for the Global Terrestrial Water Cycle) under NASA's Making Earth Science data record for Use in Research Environments (MEaSUREs) program, and NNX09AK35G (Development and diagnostic analysis of a multi-decadal global evaporation product for NEWS) under the NASA Energy and Water System (NEWS) program. The support from these programs is highly appreciated.

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



**Table 1. Summary of the gridded data used in this study**

**\*The study period is 1984 – 2010**

**\*CLM and NOAH in grey are analyzed but merged into the final water budget CDR in this study**

| Dataset | | Period | Spatial Resolution | Temporal Resolution | References |
|---|---|---|---|---|---|
| | | | Precipitation | | |
| CSU | | 1998-2010 | 0.25° | 3 h | Bytheway et al. (2013) |
| PGF | | 1948-2010 | 0.25° | 3 h | Sheffield et al. (2006) |
| CHIRPS | | 1981-present | 0.5° | monthly | Funk et al. (2014) |
| GPCC(v6) | | 1901-2010 | 0.5° | monthly | Schneider et al.(2014) |
| | | | Evapotranspiration | | |
| VIC | | 1948-2010 | 0.25° | 3 h | Sheffield and Wood (2007) |
| CLM | | 1948-2010 | 1° | 3 h | - |
| NOAH | | 1948-2010 | 1° | 3 h | - |
| ERA-Interim | | 1979-prensent | T255 | - | Simmons et al. (2006) |
| MERRA | | 1979-prensent | 2/3*1/2 H grids | - | Rienecker et al. (2011) |
| GLEAM(V2A) | | 1981-2011 | 0.25° | daily | Miraless et al. (2011) |
| SRB -PGF | PM | 1984-2010 | 0.5° | daily | Vinukollu et al. (2011) |
| | PT | | | | |
| SRB-CFSR | PM | 1984-2007 | 0.5° | daily | Vinukollu et al. (2011) |
| | PT | | | | |
| | | | Runoff | | |
| VIC | | 1948-2010 | 0.25° | 3 h | Sheffield and Wood (2007) |
| CLM | | 1948-2010 | 1° | 3 h | - |
| NOAH | | 1948-2010 | 1° | 3 h | - |
| | | | Total Water Storage (TWS) | | |
| VIC | | 1948-2010 | 0.25° | 3 h | Sheffield and Wood (2007) |
| CLM | | 1948-2010 | 1° | 3 h | - |
| NOAH | | 1948-2010 | 1° | 3 h | - |
| GRACE | | 2002-present | 1° | monthly | Landerer et al. (2012) |





**Table 2. Data sources of merged water budgets with their averaged merging weights in brackets throughout different sub-periods**

\*$TWSC$ from GRACE in 2002 is uncompleted that GRACE for 2002 is excluded

\* The spatial maps of merging weights over the globe can be found in Figure S1 to S3 in Supplement II

|  | 1984-1997 | 1998-2002 | 2003-2007 | 2008-2010 |
|---|---|---|---|---|
| P | PGF (29.6%), GPCC (34.6%), CHIRPS (35.8%) | PGF (20.7%), GPCC (26.8%), CHIRPS (26.5%), CSU (26.0%), | | |
| ET | VIC (11.3%), ERA-Interim (10.8%), MERRA (6.6%), GLEAM (12.8%), SRB-PGF-PM (17.2%), SRB-PGF-PT (15.9%), SRB-CFSR-PM (13.9), SRB-CFSR-PT (11.5%) | | | VIC (17.4%), ERA-Interim (19.6%), MERRA (11.4%), GLEAM (20.3%), SRB-PGF-PM (18.0%), SRB-PGF-PT (13.2%) |
| R | VIC | | | |
| TWSC | VIC | | VIC (67.1%), GRACE (32.9%) | |

Table 3. Annual mean water budgets (mm/year) over the globe (Greenland and Antarctica excluded) before (normal font) /after (in bold) water balance constraint, and their attributions (in italic) to non-closure error throughout sub-periods

|  | 1984-1997 | 1998-2002 | 2003-2007 | 2008-2010 | 1984-2010 |
|---|---|---|---|---|---|
| **P** | 767.0<br>**776.0**<br>*42.9%* | 792.7<br>**798.0**<br>*35.3%* | 786.7<br>**779.1**<br>*34.5%* | 802.9<br>**787.0**<br>*29.4%* | 779.4<br>**781.8**<br>*38.4%* |
| **ET** | 518.0<br>**464.7**<br>*42.0%* | 523.6<br>**467.0**<br>*47.7%* | 516.0<br>**457.6**<br>*48.2%* | 522.0<br>**465.2**<br>*53.0%* | 519.1<br>**463.9**<br>*45.4%* |
| **R** | 333.0<br>**312.1**<br>*4.6%* | 352.7<br>**330.5**<br>*5.2%* | 343.8<br>**322.0**<br>*5.2%* | 335.3<br>**318.0**<br>*5.2%* | 338.9<br>**318.0**<br>*4.9%* |
| **TWSC** | -3.4<br>**-0.8**<br>*10.5%* | -1.7<br>**0.6**<br>*11.9%* | -2.4<br>**-0.5**<br>*12.1%* | 0.9<br>**3.8**<br>*12.4%* | -2.4<br>**0.0**<br>*11.2%* |
| **Closure** | -80.6 | -82.0 | -70.8 | -55.3 | -76.2 |

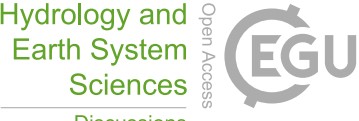



**Table 4.** The summary table of annual mean water budgets (mm/year) before and after water balance constraint and their corresponding attributions (%) to the non-closure error at both continental and basin scales (Greenland is excluded for North America)

| | water budget terms before constrain | | | | | water budget terms after constrain | | | | Attribution (%) | | | |
|---|---|---|---|---|---|---|---|---|---|---|---|---|---|
| | **P** | **ET** | **R** | **TWSC** | **Closure** | **P** | **ET** | **R** | **TWSC** | **P** | **ET** | **R** | **TWSC** |
| **Africa** | 650 | 524 | 179 | -6 | -48 | 652 | 478 | 174 | 0 | 37 | 50 | 3 | 10 |
| **Asia** | 685 | 409 | 346 | 2 | -91 | 686 | 359 | 327 | 0 | 37 | 45 | 5 | 12 |
| **Europe** | 676 | 410 | 276 | 4 | -18 | 653 | 395 | 258 | 0 | 36 | 41 | 7 | 14 |
| **NorthAmer** | 686 | 461 | 318 | 1 | -95 | 679 | 395 | 284 | 0 | 37 | 42 | 6 | 12 |
| **SouthAmer** | 1555 | 958 | 685 | -11 | -77 | 1575 | 907 | 668 | 0 | 42 | 40 | 7 | 11 |
| **Oceania** | 450 | 404 | 129 | -6 | -76 | 468 | 346 | 122 | 0 | 41 | 46 | 2 | 10 |
| **Amazon** | 2160 | 1173 | 1048 | -14 | -46 | 2182 | 1153 | 1029 | 0 | 46 | 33 | 9 | 12 |
| **Amur** | 405 | 325 | 135 | -4 | -52 | 424 | 295 | 129 | 0 | 36 | 52 | 4 | 8 |
| **Aral** | 239 | 235 | 75 | 0 | -71 | 266 | 197 | 69 | 0 | 40 | 45 | 2 | 13 |
| **Columbia** | 581 | 416 | 318 | -1 | -152 | 624 | 331 | 293 | 0 | 37 | 42 | 7 | 14 |
| **Congo** | 1454 | 1063 | 407 | -16 | 0 | 1449 | 1045 | 404 | 0 | 46 | 37 | 6 | 11 |
| **Danube** | 771 | 505 | 272 | -2 | -3 | 768 | 503 | 265 | 0 | 42 | 38 | 6 | 13 |
| **Dnieper** | 605 | 429 | 117 | 13 | 47 | 577 | 461 | 115 | 0 | 35 | 44 | 4 | 17 |
| **Don** | 514 | 402 | 96 | 19 | -3 | 498 | 404 | 94 | 0 | 42 | 39 | 3 | 16 |
| **Indigirka** | 240 | 173 | 132 | 0 | -65 | 258 | 138 | 120 | 0 | 30 | 50 | 4 | 17 |
| **Indus** | 388 | 338 | 154 | -11 | -94 | 425 | 277 | 148 | 0 | 41 | 46 | 3 | 11 |
| **Kolyma** | 276 | 194 | 125 | 3 | -46 | 283 | 167 | 116 | 0 | 29 | 48 | 4 | 18 |
| **Lena** | 366 | 267 | 142 | -7 | -36 | 379 | 245 | 134 | 0 | 30 | 51 | 4 | 15 |
| **Limpo** | 506 | 571 | 42 | -5 | -103 | 537 | 496 | 41 | 0 | 38 | 54 | 1 | 8 |
| **Mackenzie** | 388 | 293 | 189 | 10 | -105 | 413 | 241 | 173 | 0 | 32 | 49 | 6 | 13 |
| **Mekong** | 1496 | 955 | 655 | -8 | -106 | 1518 | 883 | 634 | 0 | 39 | 39 | 7 | 15 |
| **Mississippi** | 789 | 597 | 220 | -11 | -16 | 792 | 577 | 215 | 0 | 42 | 42 | 5 | 10 |
| **Murray-Darling** | 439 | 440 | 42 | -3 | -40 | 452 | 411 | 41 | 0 | 41 | 49 | 1 | 8 |
| **Niger** | 595 | 426 | 198 | -1 | -28 | 595 | 401 | 194 | 0 | 35 | 49 | 3 | 13 |
| **Nile** | 522 | 464 | 97 | -3 | -36 | 521 | 426 | 96 | 0 | 32 | 54 | 2 | 12 |
| **NoethernDvina** | 598 | 355 | 318 | 16 | -91 | 618 | 324 | 294 | 0 | 32 | 43 | 10 | 16 |
| **Ob** | 468 | 340 | 159 | 14 | -45 | 470 | 323 | 147 | 0 | 34 | 45 | 5 | 17 |
| **Olenek** | 284 | 177 | 114 | 1 | -8 | 280 | 174 | 106 | 0 | 28 | 47 | 4 | 21 |
| **Parana** | 1201 | 893 | 278 | -17 | 47 | 1171 | 892 | 279 | 0 | 39 | 47 | 4 | 10 |
| **Pearl** | 1425 | 794 | 747 | 9 | -126 | 1438 | 732 | 706 | 0 | 38 | 41 | 10 | 10 |
| **Pechora** | 563 | 246 | 342 | 57 | -82 | 552 | 244 | 308 | 0 | 33 | 38 | 8 | 20 |
| **Senegal** | 283 | 238 | 57 | -5 | -7 | 274 | 218 | 55 | 0 | 37 | 50 | 1 | 10 |
| **Ural** | 303 | 286 | 59 | -6 | -36 | 317 | 260 | 57 | 0 | 35 | 44 | 3 | 18 |
| **Volga** | 560 | 391 | 196 | 3 | -30 | 563 | 375 | 188 | 0 | 36 | 40 | 7 | 17 |
| **Yangtze** | 1017 | 623 | 522 | -5 | -123 | 1061 | 554 | 507 | 0 | 37 | 44 | 9 | 9 |
| **Yellow** | 416 | 372 | 99 | -1 | -54 | 431 | 336 | 95 | 0 | 36 | 51 | 4 | 9 |
| **Yenisei** | 437 | 300 | 217 | 22 | -102 | 460 | 265 | 195 | 0 | 31 | 48 | 6 | 15 |
| **Yukon** | 307 | 222 | 149 | 7 | -71 | 314 | 175 | 139 | 0 | 39 | 43 | 4 | 14 |





Table 5. Flux towers information list. From left to right the station name; data available time span; latitude; longitude; International Geosphere-Biosphere International Program (IGBP) land cover (Loveland et al. 2000)

| Name | Availabile Years | | | Lat | Lon | VegeType |
|---|---|---|---|---|---|---|
| DE-Geb | 2004 | - | 2006 | 51.1 | 10.91 | CRO |
| DE-Kli | 2004 | - | 2006 | 50.89 | 13.52 | CRO |
| FR-Lam | 2005 | - | 2005 | 43.49 | 1.24 | CRO |
| IT-Bci | 2004 | - | 2006 | 40.52 | 14.96 | CRO |
| US-ARM | 2003 | - | 2006 | 36.61 | -97.49 | CRO |
| US-Bo1 | 1997 | - | 2006 | 40.01 | -88.29 | CRO |
| US-Bo2 | 2004 | - | 2006 | 40.01 | -88.29 | CRO |
| IT-Noe | 2004 | - | 2006 | 40.61 | 8.15 | CSH |
| CA-Oas | 1997 | - | 2005 | 53.63 | -106.2 | DBF |
| DE-Hai | 2000 | - | 2006 | 51.08 | 10.45 | DBF |
| IT-Col | 1996 | - | 2005 | 41.85 | 13.59 | DBF |
| IT-Ro1 | 2000 | - | 2006 | 42.41 | 11.93 | DBF |
| US-MMS | 1999 | - | 2005 | 39.32 | -86.41 | DBF |
| US-Moz | 2004 | - | 2006 | 38.74 | -92.2 | DBF |
| US-WCr | 1999 | - | 2006 | 45.81 | -90.08 | DBF |
| FR-Pue | 2000 | - | 2006 | 43.74 | 3.6 | EBF |
| CA-Ca1 | 1998 | - | 2005 | 49.87 | -125.33 | ENF |
| CA-Obs | 1999 | - | 2005 | 53.99 | -105.12 | ENF |
| CA-Ojp | 1999 | - | 2005 | 53.92 | -104.69 | ENF |
| CA-Qcu | 2002 | - | 2006 | 49.27 | -74.04 | ENF |
| CA-Qfo | 2004 | - | 2006 | 49.69 | -74.34 | ENF |
| DE-Tha | 1997 | - | 2006 | 50.96 | 13.57 | ENF |
| DE-Wet | 2002 | - | 2006 | 50.45 | 11.46 | ENF |
| FR-LB | 1997 | - | 2006 | 44.72 | -0.77 | ENF |
| IL-Yat | 2001 | - | 2006 | 31.34 | 35.05 | ENF |
| IT-Lav | 2001 | - | 2006 | 45.96 | 11.28 | ENF |
| NL-Loo | 1997 | - | 2006 | 52.17 | 5.74 | ENF |
| RU-Fyo | 1998 | - | 2006 | 56.46 | 32.92 | ENF |
| SE-Nor | 1996 | - | 2005 | 60.09 | 17.48 | ENF |
| US-NR1 | 1999 | - | 2003 | 40.03 | -105.55 | ENF |
| US-Wrc | 1998 | - | 2006 | 45.82 | -121.95 | ENF |
| DE-Meh | 2004 | - | 2006 | 51.28 | 10.66 | GRA |
| ES-VDA | 2004 | - | 2006 | 42.15 | 1.45 | GRA |
| IT-Mbo | 2003 | - | 2006 | 46.02 | 11.05 | GRA |
| NL-Ca1 | 2003 | - | 2006 | 51.97 | 4.93 | GRA |
| PT-Mi2 | 2005 | - | 2006 | 38.48 | -8.02 | GRA |
| US-Aud | 2003 | - | 2006 | 31.59 | -110.51 | GRA |
| US-Bkg | 2004 | - | 2006 | 44.35 | -96.84 | GRA |
| US-CaV | 2004 | - | 2004 | 39.06 | -79.42 | GRA |
| US-Fpe | 2000 | - | 2006 | 48.31 | -105.1 | GRA |
| US-Goo | 2002 | - | 2006 | 34.25 | -89.87 | GRA |
| US-Wkg | 2004 | - | 2006 | 31.74 | -109.94 | GRA |
| JP-Tom | 2001 | - | 2003 | 42.74 | 141.51 | MF |
| CA-Mer | 1998 | - | 2005 | 45.41 | -75.52 | WET |
| CN-Do2 | 2005 | - | 2005 | 31.58 | 121.9 | WET |





| BW-Ma1 | 2000 | - | 2001 | -19.92 | 23.56 | WSA |
| US-SRM | 2004 | - | 2006 | 31.82 | -110.87 | WSA |





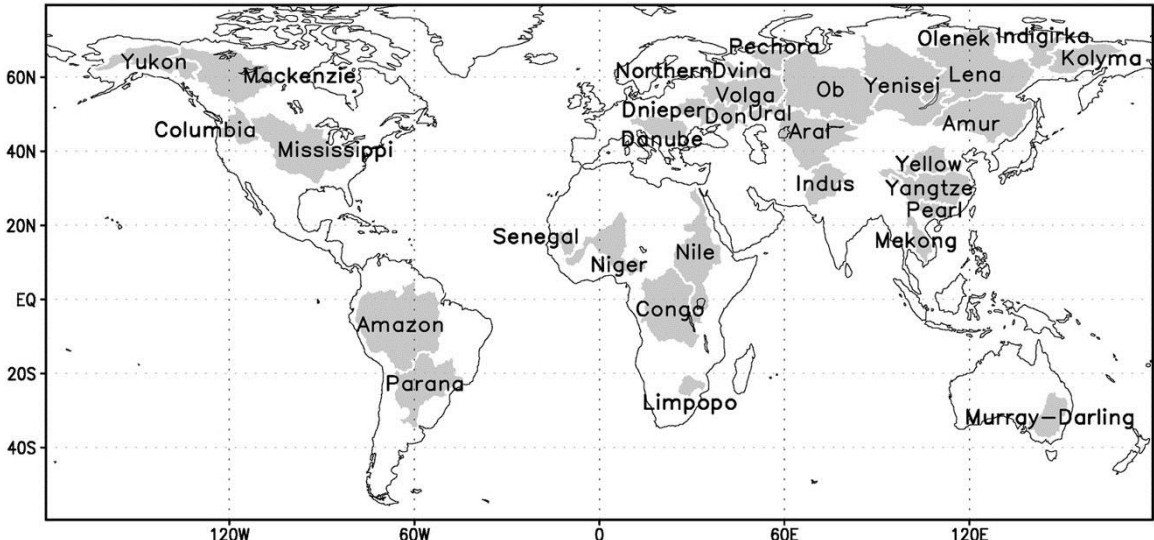

**Figure 1. Locations of 32 selected large basins (Pan et al. (2012))**




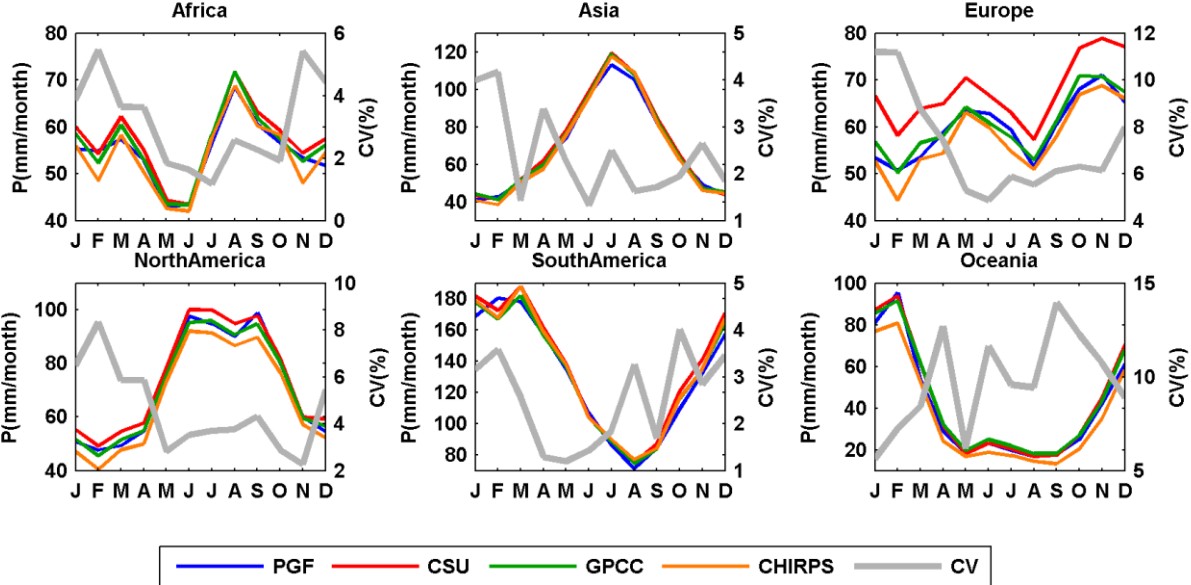

**Figure 2. Seasonal cycles of precipitation from different products over the six continents for 1998-2010**

*CHIRPS and TMPART only cover the region between 50°N-50°S; therefore, only the grids between 50°N-50°S are counted into
the calculation of the seasonal cycle

*The Coefficient of Variance (CV, %) is calculated as the standard deviation divided by the ensemble mean of all the products (The
same for Figures 3-9)





**Figure 3. Seasonal cycles of precipitation from different products over twelve representative large basins for 1998-2010**

**\*CHIRPS and CSU only cover the region between 50°N-50°S. For those basins either outside or across 50°N-50°S, only PGF and GPCC are visualized**





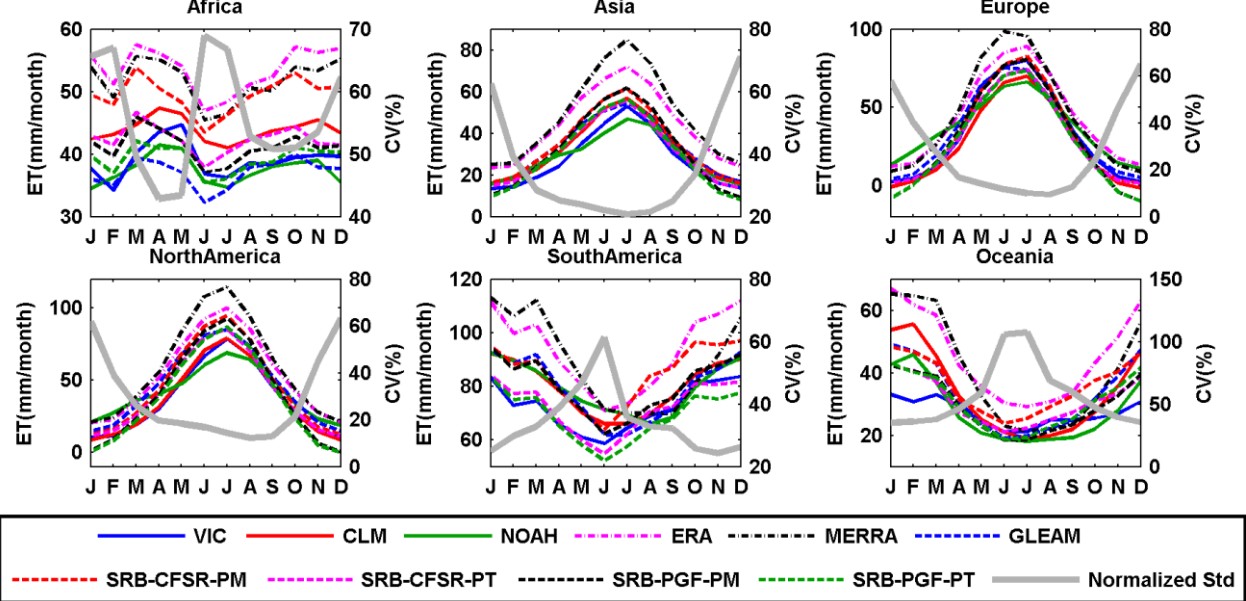

**Figure 4. Seasonal cycles of evapotranspiration from different products over the six continents for 1984-2007**

***Greenland is excluded for North America, the same for Figures 6 and 8**







**Figure 5. Seasonal cycles of evapotranspiration from different products over twelve representative large basins for 1984-2007**





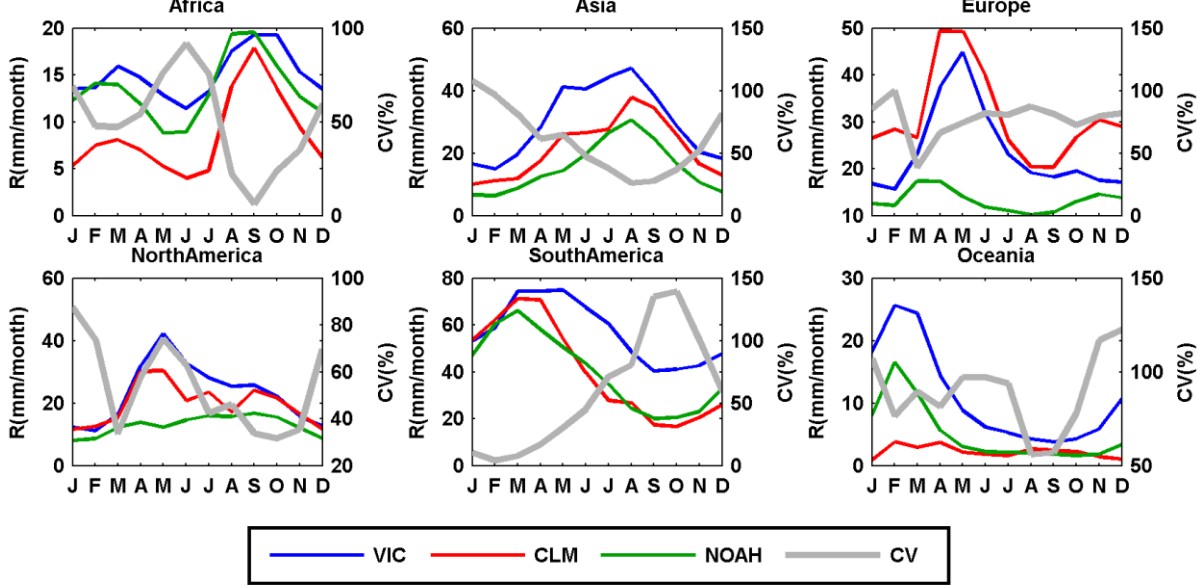

**Figure 6. Seasonal cycles of runoff from different products over the six continents for 1984-2010**





**Figure 7. Seasonal cycles of runoff from different products over twelve representative large basins for 1984-2010**




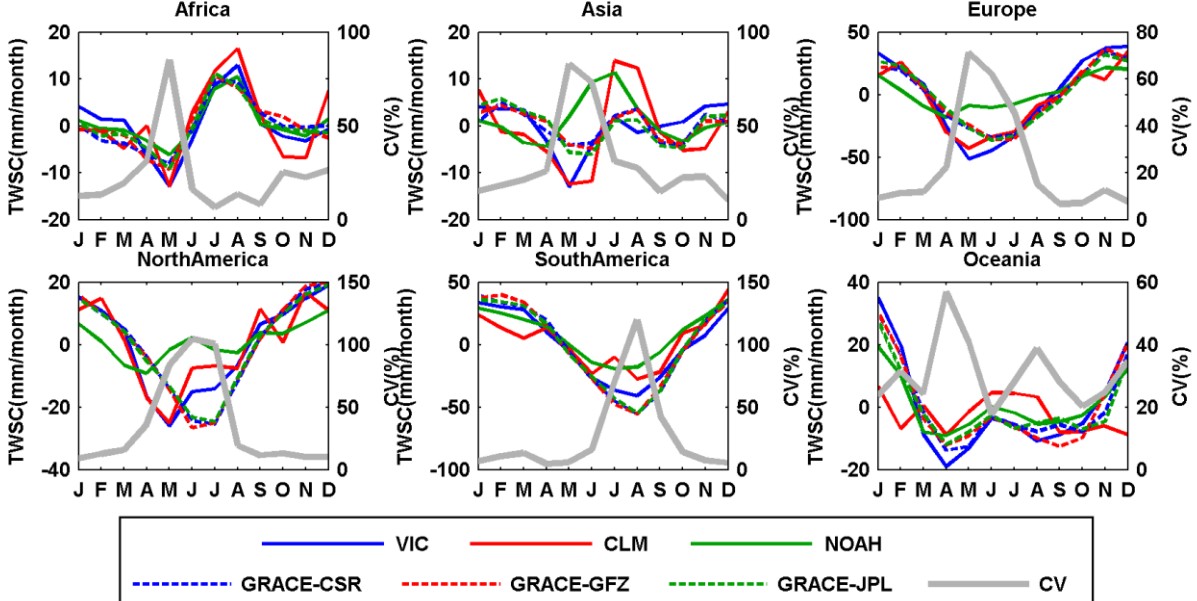

**Figure 8. Seasonal cycles of total water storage change (TWSC) from different products over the six continents for 2003-2010**

**\*$TWSC$ is first normalized and then the CV (%) is calculated (The same for Figure 9)**





**Figure 9. Seasonal cycle of total water storage change (TWSC) from different products over twelve representative large basins for 2003-2010**





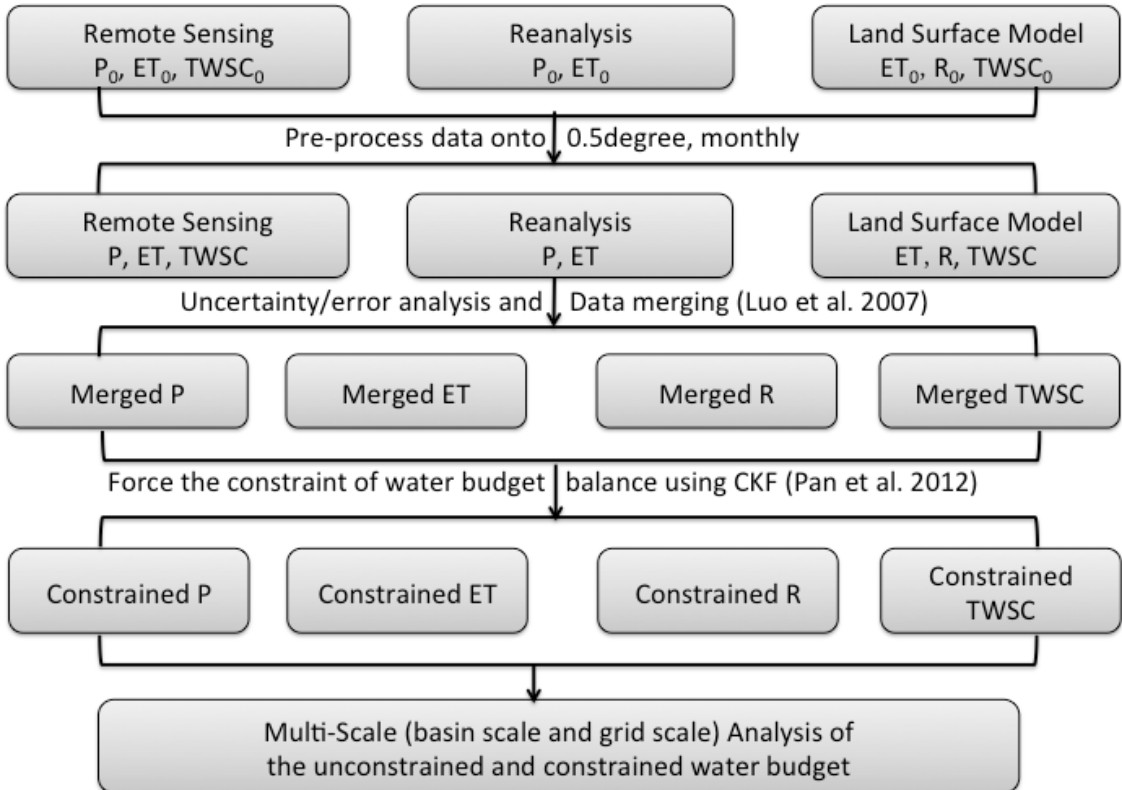

**Figure 10. Flowchart**





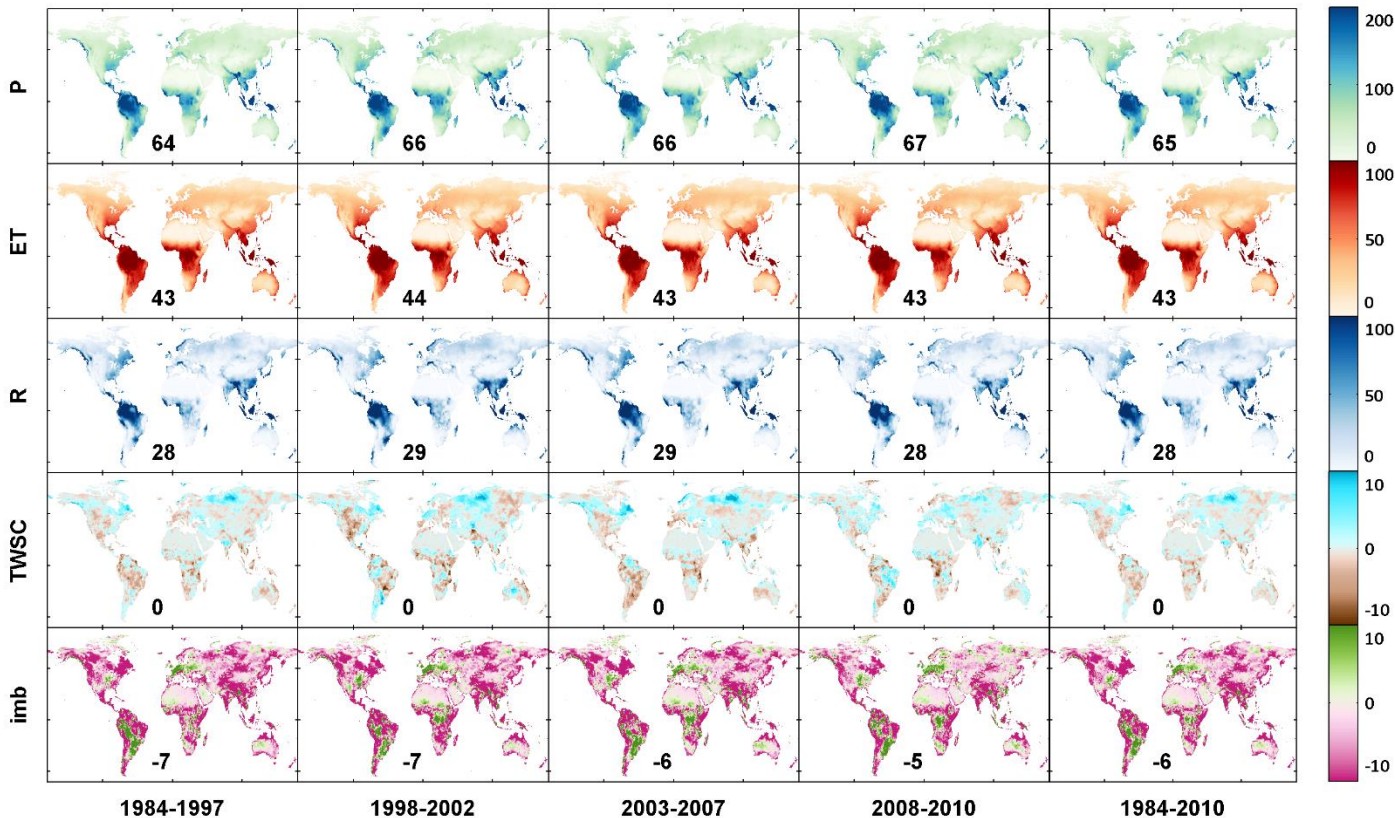

**Figure 11. Monthly mean (mm/month) of different water budget terms (from the first row to the bottom: precipitation, evapotranspiration, runoff, total water storage change, imbalance) before water balance constraint throughout different periods (from the left to the right: 1984-1997, 1998-2002, 2003-2007, 2008-2010, and 1984-2010)**

5    **\*The numbers listed on each sub-panel are the monthly mean value for each merged water budget variable before water balance constraint (mm/month) during different sub-periods (Greenland and Antarctica excluded). So as in Figures S4 and S5 in Supplementary II, but for the merged water budget variable after water balance constraint (mm/month), and the non-closure error attributions to each water budget variable (%)**





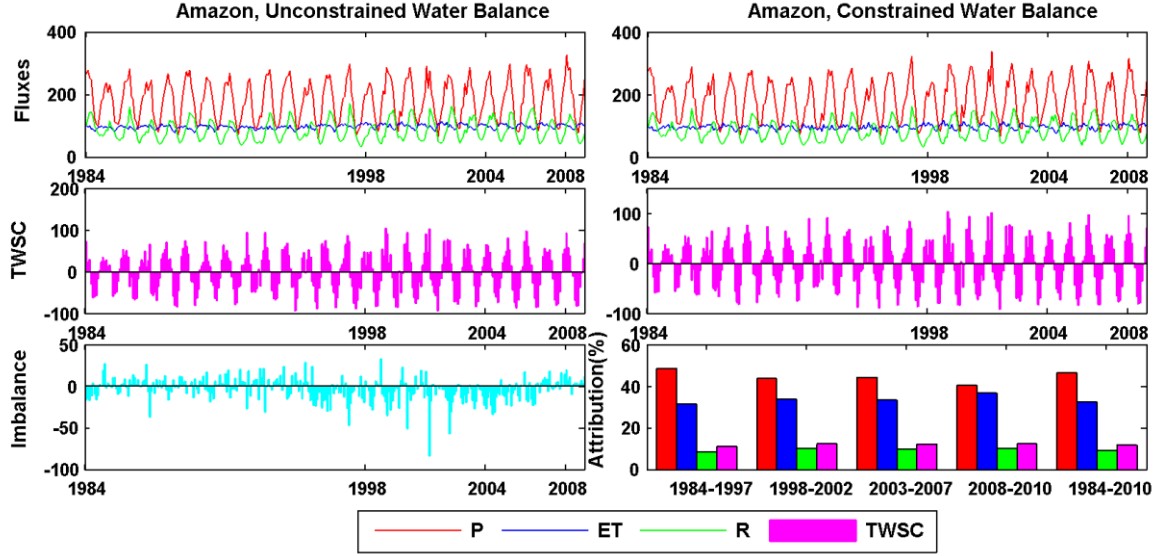

**Figure 12. Unconstrained (left column) and constrained (right column) water budget estimates (mm/month) over Amazon River basin. The top, middle and bottom rows show the time series of water budget in terms of fluxes (precipitation, evapotranspiration and runoff), $TWSC$ (Total Water Storage Change) and Imbalance. The imbalance/non-closure error after water budget constraint**
5 **equals to zero and the imbalance/non-closure attributions to each water budget variables throughout different sub-periods are shown at the right bottom.**






**Figure 13 Seasonal Cycles of runoff from VIC, CLM, NOAH and MEaSUREs against GRDC runoff observation over 26 large basins for different periods according to in situ data availability**





**Figure 14 (a)** Correlation Coefficient (CC) between monthly GRDC runoff observations and MEaSUREs runoff estimates for 165 medium basins; **(b)** same as (a), but for 862 small basins; **(c)** Monthly mean of MEaSUREs runoff estimates against GRDC runoff observations for medium basins; **(d)** same as (c), but for small basins

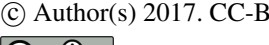



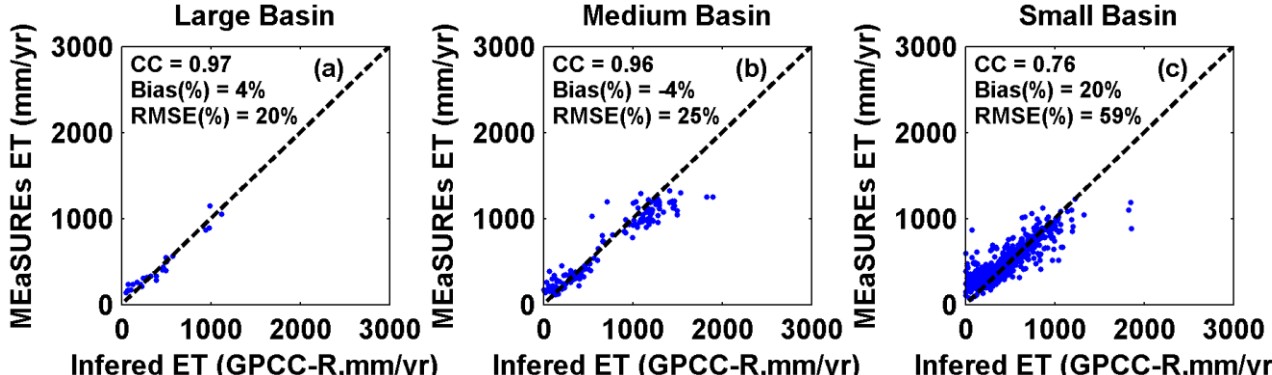

**Figure 15. Validation of MEaSUREs ET estimates against Inferred ET ($P - R$) over large (25), medium (169) and small (813) basins**

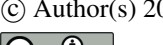



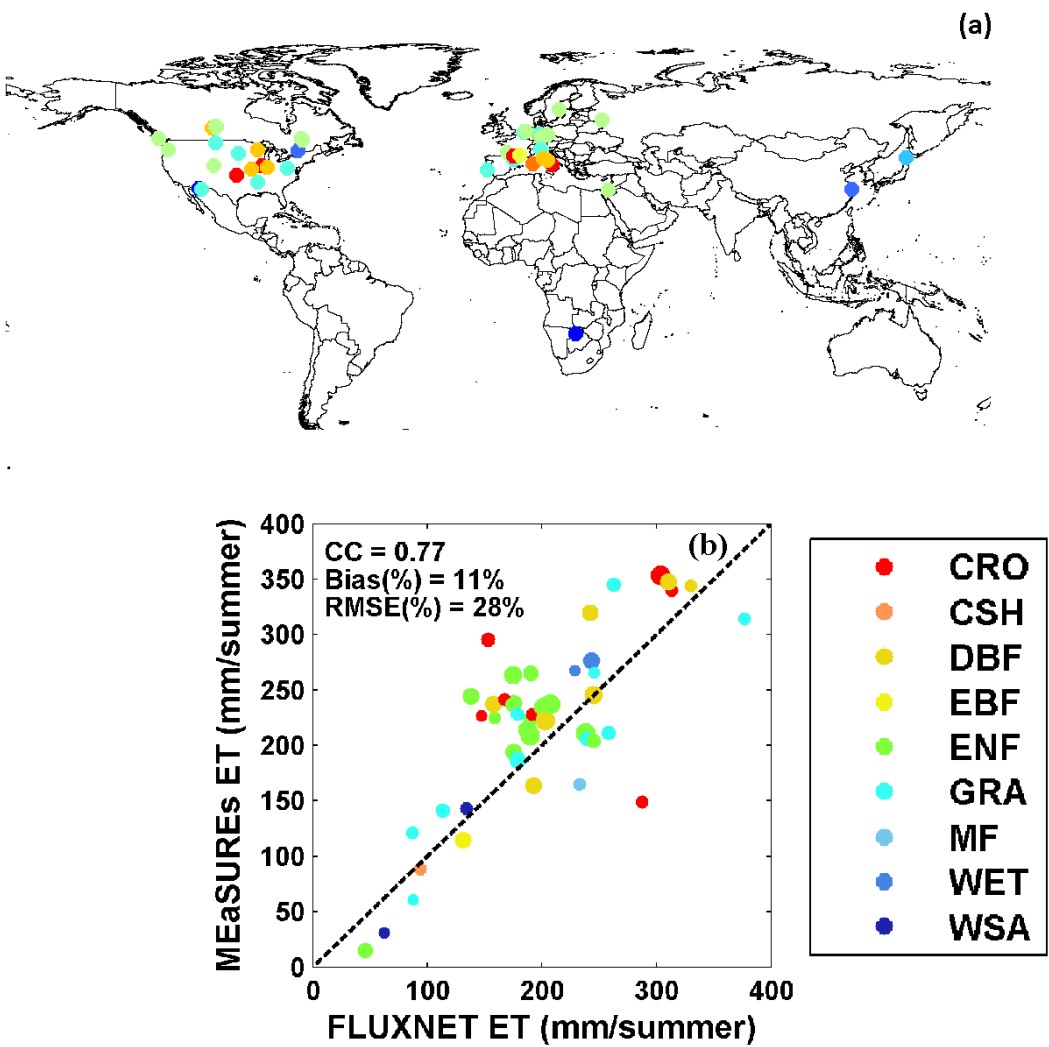

**Figure 16. (a) Distributions of 47 flux towers over the globe (b) Validation of MEaSUREs ET estimates against FLUXNET observations**

\* **Different colors in Figure 16(b) represent different International Geosphere-Biosphere International Program (IGBP) land cover types (Loveland et al. 2000); the sizes of dots represents the data record length (ranging from 1 to 10 years) from the FluxNet**

\* **IGBP land cover types: (1)CRO: Cropland; (2) CSH: Closed Shrublands; (3) DBF: Deciduous Broadleaf Forests; (4) EBF: Evergreen Broadleaf Forest; (5) ENF: Evergreen Needleleaf Forest; (6) GRA: Grassland; (7) MF: Mixed Forest; (8) WET: Permanent Wetland; (9) WSA: Woody Savannas**