# Peer review of "A Climate Data Record (CDR) for the global terrestrial water budget: 1984-2010"

_Hydrology and Earth System Sciences, 2017_

## Referee Comment (RC1) · Anonymous Referee #1 · 20 May 2017

In this manuscript the authors describe the development of a global terrestrial water budget time series at 0.5 degree spatial resolution and spanning more than two decades at monthly intervals. This work expands upon previous efforts by some of the co-authors and represents an important next step in bringing together a variety of data sources at the global scale while addressing the problem of water budget closure at the grid scale. The authors describe a rather comprehensive consideration of precipitation, evapotranspiration, runoff, and storage change datasets and how the variations in dataset extent and consistency are addressed to produce merged spatial time-series for each budget component. They then describe the grid-scale water budget constrained data assimilation process and results, notably including a presentation and discussion of attribution of closure errors. Finally, the authors present comparisons of the derived product against independent observations, noting overall adequate agreement but regions of poorer match and potential reasons for this. Overall the manuscript is well-written and describes the process and results well. Figures are consistent, descriptive, and illustrate important points from the text.

General comments: The authors propose the developed dataset as a publicly-available reference for understanding climate variability and trends. The bulk of this manuscript describes how the derived data product captures mean behavior of the terrestrial hydrology. While this is fundamental and important, less attention is paid to the extent to which the data product captures inter-annual variability. Additional text and perhaps a figure describing the climatologically-relevant variability or cycles in the produced dataset would greatly improve the manuscript and be valuable to users of the product.

This manuscript describes a temporally and spatially disaggregated global terrestrial water budget. It would be helpful to put the results of this work into the broader context of global water budget quantification by including comparisons of the derived average water budget components with previously published global budgets (e.g. Trenberth et al, 2007).

Specific comments:

Page 1 - Abstract: The method used is described as 'optimal' or 'optimizing'. Provide further explanation in the body regarding what is meant by 'optimal' in the process used.

Page 2, Line 14: The authors describe the product developed as a climate data record (CDR) that is defined as "a time series of measurements of sufficient length, consistency and continuity to determine climate variability and change". It is not necessarily clear from the description given, though, the extent to which the produced global hydrologic budget product meets this standard. Additional explanation should be included as to the nature and validity of variability and change captured in the data product.

Page 4, line 9: The authors list accounting for the Earth's oblateness as one of the

advances in this study. It is not apparent from the rest of the manuscript what precisely this refers to. Is this in reference to the use of a geographic coordinate system rather than a regular square grid? Please include a short description, where relevant, that specifies what is meant by this.

Page 5, lines 19-22: Given the proportion of land mass in Europe, Asia, and North America that exists poleward of 50 degrees N latitude, do you expect that using datasets that do not extend beyond that latitude might also account for part of the variation seen for those continents and river basins?

Page 5, lines 22-25: The spread among seasonal precipitation values for the Danube and Mississippi appears larger than that shown in Pan et al, 2012 as a result of the inclusion of the CSU dataset. Do the same potential explanations apply here, and specifically to the CSU dataset, i.e. a more dense gauge network can lead to more variability in resulting product as a result of variable application of undercatch adjustments and gridding procedures. It would be helpful to include a brief note explaining this along with the discussion of PGF, GPCC, and CHIRPS.

Page 6, lines 15-18: What additional information or value do the cross-combined SRB-CFSR, SRB-PGF and PM/PT datasets bring to the overall analysis and assimilation? In other words, what aspects of ET quantification do these combinations of algorithms provide or cover that are not addressed in the other 6 datasets? A brief justification would help clarify this point.

Page 8, Line 11: Does the resampling in space and time introduce additional error or imposed correlation that warrants treatment in the merging and data assimilation process?

Page 9, Section 2.2.2: How was the error calculated for the runoff component? Were all three sources (VIC, CLM, NOAH) used? Please clarify.

Page 10, Section 3.2 and Figure 12: The description of the example water budget con-

strained assimilation for the Amazon suggests that the precipitation component for the assimilation received the highest non-closure error attribution. If the error covariance for data assimilation is based on the spread of ensemble values for each water budget component (as described in Section 2.2.2), which appears comparatively low (∼10-20 mm for precipitation compared to >30 mm for ET, based on plots in Figures 3 & 5, respectively), how does this translate to the attributions reported? Perhaps this is obscured by the fact that the plots in Figures 3 and 5 are seasonal averages whereas the water budget closure assimilation is done monthly? Some additional explanation here (Section 3.2) or in the brief Section 2.2.2 would help clarify these sorts of apparent inconsistencies and guide the reader through the process.

Page 11 - Line 9-11: Given that human activity can impact long term water storage (multi-decadal groundwater storage decline, filling or removal of dams and reservoirs, etc), it seems that a long-term mean TWSC might not be appropriate in some locations. This assumption needs additional justification.

Additionally, how do the authors reconcile the assumption of a long-term zero trend in terrestrial water storage with studies that indicate recent trends in continental water storage (e.g. Reager et al 2016 - 'A decade of sea-level rise slowed by climate-driven hydrology' Science)?

Page 13 - Line 6: What is meant by "non-significant correlations" here? What portion of the total was filtered out for comparison?

Page 12-13, Section 4.1: The authors refer to the developed dataset alternatively as the CDR and the 'MEaSUREs' dataset within this section. Consider revising for consistency and clarity.

The comparison of the developed data product runoff against available gage records (Figures 13 & 14) indicates poorer matches in northern regions and in more arid regions. The authors describe potential reasons for the mismatch in northern basins (lake/wetland influences) and the arid southern Africa data points (poor representation

of sporadic rainfall and quick runoff). I'm curious if the poorer match in arid and semi-arid regions is potentially attributable to unaccounted-for water management activities (which tend to be more pervasive in water-limited regions) or if there is an underlying hydrologic bias specific to those areas.

Page 16, Line 1: It seems the runoff and TWSC components of this process could be improved to better represent lake/wetland dynamics which are noted as potential aspects of budget mismatch in certain regions. Do future plans entail addressing these issues?

Minor edits: Page 4, line 20: Check the tense(s)

Page 6, line 15: 'These four products are referred [to] as...' Page 6, lines 15-18: Check sentence for extra words/order

Page 14, line 4-5: Sentence wording a little unclear - consider revising for clarity

Page 20, Table 1: *CLM and NOAH in grey are analyzed but [NOT] merged into ... ERA-Interim & MERRA lines - 1979-present (misspelling)

Page 22 - Table 4: Typo - NoethernDvina -> Northern Dvina

Page 39 - Figure 15: Misspelling on plot axis: 'infered' -> 'inferred'

---

## Referee Comment (RC2) · Anonymous Referee #2 · 30 May 2017

Zhang et al. describe the development of a new climate data record that provides monthly values of precipitation, evapotranspiration, runoff and total water storage changes at 0.5 degree resolution globally from 1984-2010.  Their approach combines a variety of remote sensing, reanalysis and land surface model products using a weighting scheme based on the variance of each data source from the ensemble mean.  Water budget closure is enforced using a constrained Kalman filter to attribute the sources of budget imbalance to individual water budget terms.  I think developing a complete climate data record that is internally consistent and ensures water budget closure is an important data need that would be useful for many other scientific applications, and the authors do a good job of pulling together all of the relevant global datasets. Unfortunately,  as detailed below, I have significant concerns about the approach used to ensure closure and the assumption that variability between data sources is representative of uncertainty and error.  While I acknowledge that the authors are doing the best they can with what is currently available, I am not convinced that the approach used here is sufficient to overcome these data limitations and achieve water balance closure in a meaningful way.

**General Comments:**
1.      The biggest concern I have with this approach is the reliance on the assumption that variability between data sources is a proxy for error individual products. I understand that this assumption arises from a lack of data for direct error analysis, but I still have significant concerns about its validity. At a minimum, I think the authors need to include some analysis demonstrating that the variability between approaches is similar to this error in locations where there are observations to compare to.
2.      I'm also concerned with the weightings that emerge from this assumption. On Page 8 line 22 the authors note that this is 'optimal merging weight,' but it's not specified what this is optimal with respect to.  Given that many of the data sources are not actually independent and some approaches contribute more datasets than others, this will result in a mean that is skewed toward the approaches with the most datasets regardless of how much unique information is being provided. I think a much more thorough analysis of what is redundant in the datasets is needed to identify when 'agreement' is actually indicating certainty as opposed to repetition of inputs and assumptions that arise from data limitations (i.e. greater uncertainty).
3.      The weighting is particularly problematic for the total water storage calculations which rely on VIC and GRACE.   It is assumed that the uncertainty of VIC is 5% and GRACE is 10% (Page 10 lines 17-18) and therefore when both datasets are available VIC is weighted higher than GRACE. I have concerns about using VIC at all given that it is not actually simulating deeper groundwater storage and it does not make sense to me to weight VIC higher than GRACE when GRACE is much closer to an observation of TWS than VIC is.
4.      I disagree with the de-trending adjustment to ensure zero water storage changes over the 1984-2010 period (Page 11 lines 6-15).  It's not clear to me why this assumption is necessary and in many developed locations sustained groundwater depletions over this time period have been well documented.
5.      I think that additional discussion and analysis of the impacts of human development on this approach is needed. The outputs are verified only against basins without significant human

development (e.g. excluding basins with large dams, urban or irrigated area >2% or >20% forest cover change); however, gridded values are being provided globally both in developed and undeveloped locations. The developed climate dataset does not reflect natural conditions because some of the input datasets used reflect human activities (e.g. remote sensing ET and storage losses from GRACE) while others (e.g. simulated runoff) do not. I am concerned that it's not clear in the manuscript (1) exactly what assumptions are being made about human impacts on the individual hydrologic budget terms in the calculation and (2) that the biases causes by human activities are not well understood in this approach and may be incorrectly adjusted for with the closure adjustments made with the Kalman filter.

6.      The verification datasets used here are not necessarily independent of the input datasets themselves. I suspect that for example the flux towers used here are also used to validate (and/or calibrate) many of the remote sensing and land surface models used here. While this is probably unavoidable given the limited number of global observations networks I think this should be evaluated and discussed because it's if these aren't really independent points, it's likely that performance based on these points is a best-case scenario.

7.      In my opinion, the scientific motivation and conclusions of this work do not come out clearly enough.  I think the introduction should be refocused on the strengths and weaknesses of existing datasets and the motivation for this work rather than starting with an outline of government organizations. For example, the paragraph starting on page 2 line 22 covers all of the remote sensing products as well as bias in inferred runoff and precipitation and challenges with water budget closure. I think this discussion as well as the motivation provided in the paragraph starting on Page 3 Line 25 should be expanded and should appear sooner in the introduction.

8.      Section 2 should be expanded to provide a better summary of the strengths and weaknesses of the different datasets without relying so heavily on the supplemental material (e.g. page 5 line 14 and section 2.1.2 paragraph 1).  I think it's fine to refer to the supplement for the details of these datasets but additional discussion is needed in the main text to explain to the reader the strengths and weaknesses of these approaches and why they were chosen. For example, it is important to clearly explain here the difference between satellite data, reanalysis products and land surface models including what goes into each and what assumptions they rely on before comparisons are made. Some of this information comes up in the discussion of differences but it would be helpful to outline approaches upfront first.

9.      The figures could be improved to provide more quantitative metrics of performance especially with respect to spatial and temporal variability. For example, Figure 11 maps all of the water balance components globally in a single figure for multiple time periods but each subplot is so small it's very difficult to note the connections the authors are discussing. Some cutouts or regional assessments would be useful. Also, Figures 2-9 are repetitive and I think some of these could be moved to the supplemental material or different plotting approaches could be tested to summarize this information with less figures.

**Specific Comments:**
1.      The list of satellite products page 2 line 25 would be easier to follow in table form.
2.      Page 4 lines 3:  I think before the paragraph laying out the advantages of this approach a more thorough explanation of the weaknesses of previous approaches would be helpful. For

example, the first reason given here is the expanded use of the Constrained Kalman filter; however, the current limitations of the Kalman filter have not been explained.

3.     Table 1 should clearly differentiate land surface models from remote sensing products.

4.     Page 5 lines 2-7: This is very detailed for this intro to this section. I think it would be better to keep this high level, and provide an overview of the general approach and the organization of section 2 for the reader here.

5.     Figure 2: A more detailed caption explaining the acronyms and the difference between the grey line and the colored lines is needed. Some of this is included in the * points. You should rewrite these to incorporate all of this into a single caption. This is also true of the subsequent figures, which should be adjusted accordingly.

6.     For figures 2- 9: I think it would make more sense to plot the standard deviation rather than the coefficient of variation. The CV values clearly display a seasonal pattern caused by dividing by the mean. Since this information is already provided in the colored lines in my opinion it would be easier to understand if the grey line just showed standard deviation.  This would also address the 'abnormal high spread' noted on page 5 line 25.

7.     Section 2.1.1: Some aggregated statistics of differences in total precipitation for the major basin would be helpful to quantify the overall differences between approaches.

8.     Page 6 line 12: The derivation of the other four satellite products is described but not the GLEAM dataset.

9.     Section 2.1.3:  I think this section should include a description of how runoff is calculated in each model and the strengths and weaknesses of each approach and their systematic biases.

10.     Page 7 line 6: Can you be more specific about what type of discrepancy you are referring to (i.e. a low bias)?

11.     Page 7 line 7: Can you be more specific about the type of 'disagreement' you are referring to?

12.     Figure 13 should be figure 8 since it gets referred to after Figure 7

13.     Page 7 line 14: Should be 'capture'

14.     Page 7 line 14-15: This is unclear, can you expand on the uncertainty estimates you are referring to here?

15.     Page 7 line 19: It would be helpful to define 'total water storage change' and 'total water storage anomaly' explicitly here before getting into this discussion.

16.     Page 7: Equation 2 is not necessary in my opinion since this approach wasn't used.

17.     Page 7 line 20: It would be helpful to explain what the significant differences in these three processing centers are.

18.     Page 8 Line 10: It sounds like you are using the ensemble mean of GRACE here for future TWSC analysis and not using VIC at all but I don't think this is the case.

19.     Page 9 lines 3-10:  Some demonstration of the impact of this adjustment on the time series would be helpful here given that the authors argue it is a 'key step' for temporal consistency.

20.     Page 10 lines 22-23: Globally mean TWSC may be small but this does not mean local changes are small and if the point is 0.5degree resolution I think this could be a limitation. Some discussion of spatial variability would be helpful here.

21.     Page 13 Lines 6-7: What does it mean to be 'filtering out those basins with non-significant correlations'?  This sounds like an additional step beyond the filtering for different anthropogenic impacts. What was the threshold for this filtering and how many points were filtered because of it?

22.     Page 14 lines 9-10: Even though ET is most dominant during the summer I think that the verification should not be limited to the warm season without further justification.

---

## Author Comment (AC1) · 22 Aug 2017

In this manuscript the authors describe the development of a global terrestrial water budget time series at 0.5 degree spatial resolution and spanning more than two decades at monthly intervals. This work expands upon previous efforts by some of the co-authors and represents an important next step in bringing together a variety of data sources at the global scale while addressing the problem of water budget closure at the grid scale. The authors describe a rather comprehensive consideration of precipitation, evapotranspiration, runoff, and storage change datasets and how the variations in dataset extent and consistency are addressed to produce merged spatial time-series for each budget component. They then describe the grid-scale water budget constrained data assimilation process and results, notably including a presentation and discussion of attribution of closure errors. Finally, the authors present comparisons of the derived product against independent observations, noting overall adequate agreement but regions of poorer match and potential reasons for this. Overall the manuscript is well-written and describes the process and results well. Figures are consistent, descriptive, and illustrate important points from the text.

Comment 1 (C1): General comments: The authors propose the developed dataset as a publicly-available reference for understanding climate variability and trends. The bulk of this manuscript describes how the derived data product captures mean behavior of the terrestrial hydrology. While this is fundamental and important, less attention is paid to the extent to which the data product captures inter-annual variability. Additional text and perhaps a figure describing the climatologically-relevant variability or cycles in the produced dataset would greatly improve the manuscript and be valuable to users of the product.

This manuscript describes a temporally and spatially disaggregated global terrestrial water budget. It would be helpful to put the results of this work into the broader context of global water budget quantification by including comparisons of the derived average water budget components with previously published global budgets (e.g. Trenberth et al, 2007).

Response 1 (R1): Thanks for the reviewer's comments. The authors have added a paragraph in the "discussion and future work" section to (1) compare with other studies about the global water budgets (2) describes the climate variability from the CDR. Corresponding figures (Figures S6-8) were added into supplement II to support the text. A paper is under preparation that reviews global water budget estimates from ~20 historical studies. Including these studies in the current paper would make it simply too long, so we summarize some results and put some results in a supplement.

"Currently the authors are carrying out another study in comparing the CDR water budget records against around 20 high-impacted studies, at multiple spatial scales (i.e. continental and global). This on-going study is the first attempt to gather and compare

global water budget estimates from studies as early as 1974 (i.e. Budyko 1974) to the current study in order to provide a comprehensive overview of global water budget estimates, even though the studies focused on different periods using different data sources and have different global coverage (e.g. some of them exclude Antarctica or Greenland or both). Figure S6 in supplement II gives an example comparison with (Trenberth et al., 2007, T2007 hereafter), which estimated the water budget during 1979-2000 and excluded Antarctica. The total precipiation is quite close to this study ($114 \times 10^3 km^3/yr$) to T2007 ($113 \times 10^3 km^3/yr$). By converting the water budgets into mm/yr based on the global coverage information available in each of those studies, the long-term mean precipiation is around 28 mm/yr (vs. 32 mm/yr in the CDR and 27 mm/yr from T2007), $ET$ is around 78 mm/yr (vs. 78 mm/yr in the CDR and 77 mm/yr from T2007), and runoff is around 47 mm/yr (vs. 46mm/yr in this study and T2007). Figure S7 further provides an example of how the CDR precipiation time series captured the 1998-1999 US drought. The 6-month SPIs exceeds the threshold of exceptional drought (which is defined by the US drought Monitor system; http://droughtmonitor.unl.edu/AboutUs/ClassificationScheme.aspx) around the year 1998. The CDR developed in this study, as a time series of measurements of sufficient length, consistency and continuity, can also be applied to determine climate variability. Figure S8 in the supplement II, as an example, provides the inter-annual variability of the available water ($P$-$ET$) over the globe during the CDR period 1984 -2010. ”

Specific comments:
C2: Page 1 - Abstract: The method used is described as 'optimal' or 'optimizing'. Provide further explanation in the body regarding what is meant by 'optimal' in the process used.

R2: It still remains a challenge to find a "best" approach to merge multiple data sources into a single data set over the globe due to limited observational data. This study assumes the deviation from the ensemble mean of all data sources, for the same budget variable, as a proxy for the uncertainty/error in individual products and uses this error information to produce the merged water budget variables, which we considered as an 'optimal' approach. This is also described in the beginning of section 2.2.1:
"There is no best estimate or observation of each individual water budget component at the grid scale over the globe due to the limited spatial coverage of in-situ measurements. This is especially true for evapotranspiration observations from the flux tower networks. Thus, the limited availability of gridded ground observations makes it impossible to quantify the error in each water budget component. Therefore, in this study, the deviation from the ensemble mean of all data sources for the same budget variable is used as a proxy of the uncertainty/error in individual products. The merging procedure for each budget component is a weighted averaging, where the optimal merging weight $w_i$ is given by the following equation…"

C3: Page 2, Line 14: The authors describe the product developed as a climate data record (CDR) that is defined as "a time series of measurements of sufficient length, consistency and continuity to determine climate variability and change". It is not necessarily clear from the description given, though, the extent to which the produced global hydrologic

budget product meets this standard. Additional explanation should be included as to the nature and validity of variability and change captured in the data product.

R3: We're sympathetic to the reviewer's comment, and in many ways for any CDR variable being proposed by GCOS this is an issue. Please see R1 (the response to your general comments.) We have added text regarding this issue in section 5. (Discussion and future work) that we feel addresses the comment.

C4: Page 4, line 9: The authors list accounting for the Earth's oblateness as one of the advances in this study. It is not apparent from the rest of the manuscript what precisely this refers to. Is this in reference to the use of a geographic coordinate system rather than a regular square grid? Please include a short description, where relevant, that specifies what is meant by this.

R4 It means when the spatial mean of water budget variable (e.g. the numbers listed in Figure 12) was calculated, it is not simply considered as the arithmetic mean of all the grid cell values but a weighted averaged value based on the area of each grid cell which considers the Earth's oblateness.

C5: Page 5, lines 19-22: Given the proportion of land mass in Europe, Asia, and North America that exists poleward of 50 degrees N latitude, do you expect that using datasets that do not extend beyond that latitude might also account for part of the variation seen for those continents and river basins?

R5: We tried not to mix the fully global datasets with datasets that only span 50S-50N when calculating the continental or basin averages. The data sets are at 0.5 degree spatial resolution and are averaged onto continents in Figure 2 and basins in Figure 3. For the Asia, Europe and North America that exist poleward beyond 50 degree north, the variation seen from Figure 3 are only calculated from the grids between 50N to 50S. This is clarified in the caption of Figure 3. Similarly for basins like Lena, Mackenzie and Yukon that are either above 50N or are across 50N, only the grids between 50N-50S were used. So we believe that the variation seen for those continents and basins come from the datasets that do not extend beyond 50N latitude.

C6: Page 5, lines 22-25: The spread among seasonal precipitation values for the Danube and Mississippi appears larger than that shown in Pan et al, 2012 as a result of the inclusion of the CSU dataset. Do the same potential explanations apply here, and specifically to the CSU dataset, i.e. a more dense gauge network can lead to more variability in resulting product as a result of variable application of undercatch adjustments and gridding procedures. It would be helpful to include a brief note explaining this along with the discussion of PGF, GPCC, and CHIRPS.

R6: The different, larger spread among seasonal values in the Danube (the Mississippi was not displayed in Pan et al. 2012) shown in this study relative to Pan et al. 2012 should be sourced from different data sources applied during the different time period. And yes, "a denser gauge network can also lead to more variability in the resulting product as a result of variable application of under-catch adjustments and gridding

procedures." And this is discussed in the manuscript in section 2.1.1 using as example basins the Danube and Mississippi.

"It is interesting to note that the average discrepancy between the highest estimates (CSU) and the lowest (CHIRPS) over Europe is around 15mm/month throughout the year (Figure 2). This discrepancy is more prominent at basin scales; for example, the monthly mean difference between CSU and CHIRPS in the densely gauged basins such as Danube and Mississippi is around 20 mm/month (Figure 3). CHIRPS is a blended precipitation product (e.g. precipitation climatology, remote sensing from multiple sources, seasonal forecast form Climate Forecast System Version 2 (CFSv2), and in situ observations) but it is dominated by gauge corrections in regions with higher gauge density such as Europe and North America, and therefore in basins such as the Danube and Mississippi. The differences among the three gauge-merged products PGF, GPCC and CHIRPS might possibly be from the different data sources that they merge rather than from gauge observations, different numbers of gauges used and under-catch corrections."

C7: Page 6, lines 15-18: What additional information or value do the cross-combined SRBCFSR, SRB-PGF and PM/PT datasets bring to the overall analysis and assimilation? In other words, what aspects of ET quantification do these combinations of algorithms provide or cover that are not addressed in the other 6 datasets? A brief justification would help clarify this point.

R7: Basically they expand the ensemble of algorithms. As part of the MeaSUREs satellite products, the combined SRB-CFSR, SRB-PGF and PM/PT utilize Surface Radiation Budget as an input but apply different algorithms (i.e. PM and PT) than the other satellite product GLEAM and other reanalysis and modeled products. Satellite observations require retrieval algorithms to estimate the geophysical variable, even though the observations are at fine spatial resolution and have comprehensive coverage. The retrieval algorithms make it possible to estimate the water budget in sparsely gauged regions. For ET, a number of satellite products are needed for the algorithms and were used in this study. This is also described in the text:

"As parts of the MeaSUREs products, the four other satellite products are derived using two algorithms, the Penman-Monteith (PM) and Priestly-Taylor (PT), cross-combined with two forcing inputs that are different from the other six *ET* products, the SRB-CFSR (Surface Radiation Budget – Climate Forecast System Reanalysis) and SRB-PGF. These four products are referred as: SRB-CFSR-PM, SRB-CFSR-PT, SRB-PGF-PM and SRB-PGF-PT (Vinukollu et al., 2011). Satellite remote sensing, carries the mission of observing Earth at fine spatial resolution and comprehensive coverage and makes it possible to estimate water budget in sparsely gauged regions. Therefore, 5 satellite *ET* products are merged into the CDR."

C8: Page 8, Line 11: Does the resampling in space and time introduce additional error or imposed correlation that warrants treatment in the merging and data assimilation process?

R8: Resampling in time is actually an up-scaling (aggregation) process, which sums the high temporal resolution data to the monthly scale. This should not introduce additional error, but should reduce the uncertainty. But for resampling in space, aggregation of those

high-resolution (e.g. 0.25 deg for CSU and PGF) onto 0.5 deg might smooth the spatial variability. Nevertheless, the resampling in space and time is a necessary step to organize all the data sources into a uniform spatial and temporal resolution for data assimilation.

C9: Page 9, Section 2.2.2: How was the error calculated for the runoff component? Were all three sources (VIC, CLM, NOAH) used? Please clarify.

R9: We added text in section 2.2.2 to clarify how the runoff error was assumed. "In this study, the error of runoff is simply assumed as 10%, as VIC is the single source of runoff." And we are aware that this number is highly empirical and based on the authors' knowledge and confidence about the model calibration given there is no global grid level (0.5 degree in this study) runoff observations to quantify the error.

R10: Page 10, Section 3.2 and Figure 12: The description of the example water budget constrained assimilation for the Amazon suggests that the precipitation component for the assimilation received the highest non-closure error attribution. If the error covariance for data assimilation is based on the spread of ensemble values for each water budget component (as described in Section 2.2.2), which appears comparatively low (10-20 mm for precipitation compared to >30 mm for ET, based on plots in Figures 3 & 5, respectively), how does this translate to the attributions reported? Perhaps this is obscured by the fact that the plots in Figures 3 and 5 are seasonal averages whereas the water budget closure assimilation is done monthly? Some additional explanation here (Section 3.2) or in the brief Section 2.2.2 would help clarify these sorts of apparent inconsistencies and guide the reader through the process.

R10: Yes, Figures 3 and 5 are seasonal averages but the water budget closure assimilation is done monthly. In order to avoid the intuitive "inconsistence" feeling, one sentence is added at the end of section 2.2.2 to further clarify the water budget closing procedure: "The water budget closure is done monthly based on variational error from month to month."

C11: Page 11 - Line 9-11: Given that human activity can impact long term water storage (multi-decadal groundwater storage decline, filling or removal of dams and reservoirs, etc), it seems that a long-term mean TWSC might not be appropriate in some locations. This assumption needs additional justification. Additionally, how do the authors reconcile the assumption of a long-term zero trend in terrestrial water storage with studies that indicate recent trends in continental water storage (e.g. Reager et al 2016 - 'A decade of sea-level rise slowed by climate-driven hydrology' Science)?

R11: We'd like to thank the reviewer for pointing out this. We struggled with this issue, but the data sources for water management, particularly groundwater extraction and comprehensive reservoir storage changes, are simply unavailable at this time. We agree that at both local and regional scales, some of the places have experienced groundwater depletions such as US high plains and central valley, western Iran, India, etc., starting from different years. But from the global perspective, over the ~three decades covered by

the study period 1984-2010, in light of the lack of data the authors assume the long term *TWSC* to be zero and thus apply the de-trending.

R12: Page 13 - Line 6: What is meant by "non-significant correlations" here? What portion of the total was filtered out for comparison?

R12: 33 medium basins (out of 362) and 36 small basins (out of 862) were filtered out by running a test of significance to remove those catchments with non-significant correlations between GRDC runoff observations and CDR runoff records. This was done in order to remove those basins such as Indus and Senegal which might have incorrect observational data. These observational data records that we believe to be incorrect are also discussed in the text:
"Note that the seasonal peaks from Noah and VIC are in agreement for the Indus basin but their peaks precede the peak from the GRDC observations, which strangely happen in November. Comparing to other studies for the Indus River (Bookhagen and Burbank, 2010), show that the discharge peak occurs in the summer time , which is consistent with VIC and Noah. Likewise for Senegal River, records from regional studies (Andersen et al., 2001) and (Stisen et al., 2008) show runoff peaks in August to September instead of April to May from the GRDC record."

C13: Page 12-13, Section 4.1: The authors refer to the developed dataset alternatively as the CDR and the 'MEaSUREs' dataset within this section. Consider revising for consistency and clarity. The comparison of the developed data product runoff against available gage records (Figures 13 & 14) indicates poorer matches in northern regions and in more arid regions. The authors describe potential reasons for the mismatch in northern basins (lake/wetland influences) and the arid southern Africa data points (poor representation of sporadic rainfall and quick runoff). I'm curious if the poorer match in arid and semiarid regions is potentially attributable to unaccounted-for water management activities (which tend to be more pervasive in water-limited regions) or if there is an underlying hydrologic bias specific to those areas.

R13: Thanks for the comment, and we have changed the text to use consistently CDR.
As for the poorer match in arid and semiarid regions, we don't believe they are due to water management because in the validation data sets the potential water management activities were excluded via those criteria mentioned in the first paragraph of section 4.1. "Basins under any one or more of the following conditions were excluded: (1) GRDC basins for which the catchment boundaries could not be reliably determined; (2) basins with large dams (reservoir capacity greater than 10% of annual streamflow); (3) basins with urban areas greater than 2% (using the ''artificial areas'' class of the map from GlobCover, version 2.3; (Bontemps et al., 2011)); (4) basins with irrigated areas greater than 2% (using the Global Irrigated Area Map; http://www.iwmigiam.org); and (5) basins with either a gain or loss forest (change in land cover) > 20% of the basin area. For both the medium and small basins, those basins with data records length less than 5 years were also excluded." But for those small basins, "though they were filtered in an attempt to remove basins impacted by factors such as reservoirs, irrigation, urbanization, and so forth, they might be impacted by the scaling issues."

C14: Page 16, Line 1: It seems the runoff and TWSC components of this process could be improved to better represent lake/wetland dynamics which are noted as potential aspects of budget mismatch in certain regions. Do future plans entail addressing these issues?

R14: Yes, the current TWSC term from the LSM does not include lake or wetland dynamics. Lake and wetland modeling was first introduced in VIC model for small lakes (smaller than computing pixel) with no river connection (Bowling et al., 2010) and then re-structured for large lakes with river connections (Gao et al., 2011). So far, a number of major global lakes and wetlands (Melton et al., 2013; Bohn et al., 2013) have been modeled, even though the coverage is not complete globally. The plan for the future is to include all major global lakes and all wetlands such that the storage dynamics can be better captured. This is being done under the NASA Surface Water Ocean Topography mission funding. Again, global data sets are not well organized to include these at this time, but we expect to include then in the future.

Reference:
Bohn, T. J., Podest, E., Schroeder, R., Pinto, N., McDonald, K. C., Glagolev, M., Filippov, I., Maksyutov, S., Heimann, M., Chen, X., and Lettenmaier, D. P.: Modeling the large-scale effects of surface moisture heterogeneity on wetland carbon fluxes in the West Siberian Lowland, Biogeosciences, 10, 6559-6576, https://doi.org/10.5194/bg-10-6559-2013, 2013.
Bowling L C and Lettenmaier D P 2010 Modeling the effects of lakes and wetlands on the water balance of arctic environments J. Hydrometeorol. 11 276–95
Gao, H., T.J. Bohn, E. Podest, K.C. McDonald, and D.P. Lettenmaier, 2011: On the cause of the shrinking of Lake Chad. Environ. Res. Lett. 6 034021, doi: 10.1088/1748-9326/6/3/034021
Melton, J. R., Wania, R., Hodson, E. L., Poulter, B., Ringeval, B., Spahni, R., et al. (2013). Present state of global wetland extent and wetland methane modelling: conclusions from a model intercomparison project (WETCHIMP). Biogeosciences, 10, 753-788. doi:10.5194/bg-10-753-2013.

Minor edits: Page 4, line 20: Check the tense(s)
Page 6, line 15: 'These four products are referred [to] as. . .' Page 6, lines 15-18: Check sentence for extra words/order
Page 14, line 4-5: Sentence wording a little unclear - consider revising for clarity
Page 20, Table 1: *CLM and NOAH in grey are analyzed but [NOT] merged into . . .
ERA-Interim & MERRA lines - 1979-present (misspelling)
Page 22 - Table 4: Typo - NoethernDvina -> Northern Dvina
Page 39 - Figure 15: Misspelling on plot axis: 'infered' -> 'inferred'
We'd appreciate the review's time in pointing out these detailed edits which will definitely lead to an improved manuscript. Those minor edits above have already been modified in the manuscript.

---

## Author Comment (AC2) · 22 Aug 2017

Response to Reviewer #2

Zhang et al. describe the development of a new climate data record that provides monthly values of precipitation, evapotranspiration, runoff and total water storage changes at 0.5 degree resolution globally from 1984-2010. Their approach combines a variety of remote sensing, reanalysis and land surface model products using a weighting scheme based on the variance of each data source from the ensemble mean. Water budget closure is enforced using a constrained Kalman filter to attribute the sources of budget imbalance to individual water budget terms. I think developing a complete climate data record that is internally consistent and ensures water budget closure is an important data need that would be useful for many other scientific applications, and the authors do a good job of pulling together all of the relevant global datasets. Unfortunately, as detailed below, I have significant concerns about the approach used to ensure closure and the assumption that variability between data sources is representative of uncertainty and error. While I acknowledge that the authors are doing the best they can with what is currently available, I am not convinced that the approach used here is sufficient to overcome these data limitations and achieve water balance closure in a meaningful way.

We'd like to thank the reviewer for reviewing and providing comments and suggestions that will lead to an improved manuscript. We've carefully considered the review comments and addressed each comment point-by-point. Our replies appear in blue font for the ease of reading.

**General Comments:**
C1. The biggest concern I have with this approach is the reliance on the assumption that variability between data sources is a proxy for error individual products. I understand that this assumption arises from a lack of data for direct error analysis, but I still have significant concerns about its validity. At a minimum, I think the authors need to include some analysis demonstrating that the variability between approaches is similar to this error in locations where there are observations to compare to.

R1: Finding a best approach for estimating uncertainties between different data sources over the globe still remains a big challenge given the limited observational data coverage on Earth's surface. This is also mentioned in (Tian and Peters‐Lidard, 2010), "The difficulty in assembling a globally consistent error map lies in the lack of gauge or radar coverage over most areas of Earth's surface". Following the approach that was proposed by (Adler et al. 2001) and was recently applied by (Tian and Peters‐Lidard, 2010), this study uses the variability between data sources to quantify the uncertainties/errors in each water budget variable. And the authors also compared the attribution of non-closure term for water budget variables with the study of Pan et al. (2012), and found the results are in general in agreement by using this approach (Page 11). A careful reading of the literature for other, less observed, variables like ET suggests that there may be no locations where all the water budget are sufficiently observed to meet the standards of the test suggested by the reviewer. This is one of the major challenges facing our science.

Reference:

Adler, R. F., Kidd, C., Petty, G., Morissey, M., and Goodman, H. M.: Intercomparison of global precipitation products: The third Precipitation Intercomparison Project (PIP-3), Bulletin of the American Meteorological Society, 82, 1377-1396, 2001.

Tian, Y., and Peters-Lidard, C. D.: A global map of uncertainties in satellite-based precipitation measurements, Geophysical Research Letters, 37, 2010.

Pan, M., Sahoo, A. K., Troy, T. J., Vinukollu, R. K., Sheffield, J., and Wood, E. F.: Multisource estimation of long-term terrestrial water budget for major global river basins, Journal of Climate, 25, 3191-3206, 2012.

C2. I'm also concerned with the weightings that emerge from this assumption. On Page 8 line 22 the authors note that this is 'optimal merging weight,' but it's not specified what this is optimal with respect to. Given that many of the data sources are not actually independent and some approaches contribute more datasets than others, this will result in a mean that is skewed toward the approaches with the most datasets regardless of how much unique information is being provided. I think a much more thorough analysis of what is redundant in the datasets is needed to identify when 'agreement' is actually indicating certainty as opposed to repetition of inputs and assumptions that arise from data limitations (i.e. greater uncertainty).

R2: Similarly to the reply above, estimating uncertainties between different data sources over the globe still remains a big challenge due to limited ground observations. This study tries to fuse as much information as is available to map the global uncertainties for each water budget variable, thus forcing the water balance closure via the uncertainties information. While some of the data share the same input, e.g. satellite observations, they use different algorithms to retrieve or calculate the corresponding water budget terms. It's hard to quantify how "independent" one data source is from another, or which one contributes more datasets than others over the globe, again, due to the limited coverage of observations. Therefore, the authors followed the existing approach used by (Adler et al. 2001) and (Tian and Peters‐Lidard, 2010) into this global study, though we understand this still remains a concern, as the reviewer mentions.

C3. The weighting is particularly problematic for the total water storage calculations which rely on VIC and GRACE. It is assumed that the uncertainty of VIC is 5% and GRACE is 10% (Page 10 lines 17-18) and therefore when both datasets are available VIC is weighted higher than GRACE. I have concerns about using VIC at all given that it is not actually simulating deeper groundwater storage and it does not make sense to me to weight VIC higher than GRACE when GRACE is much closer to an observation of TWS than VIC is.

R3: We recognize the concerns of the reviewer. The numbers (i.e. 5% for VIC and 10% for GRACE) that the authors applied in this study are highly empirical and based on the authors' knowledge and confidence about the model calibration. VIC has been calibrated globally against streamflow gauges. GRACE uses a complex correction algorithm and gets rescaled using the CLM land surface model which to our understanding hasn't been globally calibrated. Almost all the monthly dynamics in soil water storage occurs in the upper soil zone, which gets captured by VIC (and other land surface models). The uncertainties in *TWSC* term have rarely been studied at large scale. In particular, the GRACE actual footprint size is around 220 km, which is very coarse compared to the grid size of this study is 0.5 degree thus the authors assume a higher uncertainty in GRACE than in VIC for *TWSC*.

C4. I disagree with the de-trending adjustment to ensure zero water storage changes over the 1984-2010 period (Page 11 lines 6-15). It's not clear to me why this assumption is necessary and in many developed locations sustained groundwater depletions over this time period have been well documented.

R4: We'd like to thank the reviewer for this comment. We agree that at regional scales, some places have experienced groundwater depletions such as US high plains and central valley, western Iran, India etc., starting from different years. One of the challenges is a lack of data on groundwater extractions. But from the global perspective, for almost three decades during the study period 1984-2010 covered by this study, the authors assume the long term *TWSC* to be zero thus apply the de-trending. We have added a discussion in the revised manuscript.

"The long term mean of *TWSC* at each grid cell is zero over the entire 27 years after the second filter, which is also named as "*TWSC* de-trending". Though at regional scales, some places have experienced groundwater depletions such as US high plains and central valley, western Iran, India etc., starting from different years. One of the challenges is a lack of data on groundwater extractions. Therefore, from the global perspective, for almost three decades during the study period 1984-2010 covered by this study, the authors assume the long term *TWSC* to be zero thus apply the de-trending, after which the spatial variability of *TWSC* still exists during the four sub-periods (Figure S4)."

C5. I think that additional discussion and analysis of the impacts of human development on this approach is needed. The outputs are verified only against basins without significant human development (e.g. excluding basins with large dams, urban or irrigated area >2% or >20% forest cover change); however, gridded values are being provided globally both in developed and undeveloped locations. The developed climate dataset does not reflect natural conditions because some of the input datasets used reflect human activities (e.g. remote sensing ET and storage losses from GRACE) while others (e.g. simulated runoff) do not. I am concerned that it's not clear in the manuscript (1) exactly what assumptions are being made about human impacts on the individual hydrologic budget terms in the calculation and (2) that the biases causes by human activities are not well understood in this approach and may be incorrectly adjusted for with the closure adjustments made with the Kalman filter.

R5: The impact of human development on water budget balance is beyond the scope of this paper. But the authors do recognize the importance of these in modifying the global water cycle. While there have been regional studies (e.g. Barnett et al., 2008; Buytaert et al., 2006), and some global simulations (e.g. Wu et al., 2013), the authors would claim that the available data sets are still too incomplete to comprehensively include their effects in a comprehensive analysis. Nonetheless, the community (and the authors) are making progress (see Wada et al., 2017 for a review) and in a future paper will update the budget numbers with (hopefully) the impacts included.

Reference:
Barnett, T. P., Pierce, D. W., Hidalgo, H. G., Bonfils, C., Santer, B. D., Das, T., Bala, G., Wood, A. W., Nozawa, T., Mirin, A. A., Cayan, D. R., and Dettinger, M. D.: Human-Induced Changes in the Hydrology of the Western United States, Science, 319, 1080-1083, 10.1126/science.1152538, 2008.
Buytaert, W., Célleri, R., De Bièvre, B., Cisneros, F., Wyseure, G., Deckers, J., and Hofstede, R.: Human impact on the hydrology of the Andean páramos, Earth-Science Reviews, 79, 53-72, http://dx.doi.org/10.1016/j.earscirev.2006.06.002, 2006.

Wu, P., Christidis, N., and Stott, P.: Anthropogenic impact on Earth/'s hydrological cycle, Nature Clim. Change, 3, 807-810, 10.1038/nclimate1932

http://www.nature.com/nclimate/journal/v3/n9/abs/nclimate1932.html - supplementary-information, 2013.

Wada, Y., Bierkens, M. F. P., de Roo, A., Dirmeyer, P. A., Famiglietti, J. S., Hanasaki, N., Konar, M., Liu, J., Müller Schmied, H., Oki, T., Pokhrel, Y., Sivapalan, M., Troy, T. J., van Dijk, A. I. J. M., van Emmerik, T., Van Huijgevoort, M. H. J., Van Lanen, H. A. J., Vörösmarty, C. J., Wanders, N., and Wheater, H.: Human-water interface in hydrological modeling: Current status and future directions, Hydrol. Earth Syst. Sci. Discuss., https://doi.org/10.5194/hess-2017-248, in review, 2017.

C6. The verification datasets used here are not necessarily independent of the input datasets themselves. I suspect that for example the flux towers used here are also used to validate (and/or calibrate) many of the remote sensing and land surface models used here. While this is probably unavoidable given the limited number of global observations networks I think this should be evaluated and discussed because it's if these aren't really independent points, it's likely that performance based on these points is a best-case scenario.

R6: The reviewer's comment is valid, we fully agree with this assessment. Data developed from either satellite remote sensing or model are often calibrated against "ground truth", i.e., gauge observations, which are also the best "ground truth" that are normally used for verification. The "ground truth" is no way independent from those remote sensing or modeled data. We have added text to the discussion to address this comment.

"The CDR is validated against ground observations, i.e. GRDC, USGS and Australian Land and Water Resources Audit project for runoff and FluxNet for $ET$, which seem not independent from the merged and constrained CDR. However, data developed from either satellite remote sensing or model are often calibrated against "ground truth", i.e., gauge observations, which are also the best "ground truth" that are normally used for verification. The "ground truth" is no way independent from those remote sensing or modeled data, particularly for global data validation. Nevertheless, we believe these data records represent the best, current knowledge for the global terrestrial water budget at the $0.5^{o}$ and monthly scale over the 27-year period of 1984-2010."

C7. In my opinion, the scientific motivation and conclusions of this work do not come out clearly enough. I think the introduction should be refocused on the strengths and weaknesses of existing datasets and the motivation for this work rather than starting with an outline of government organizations. For example, the paragraph starting on page 2 line 22 covers all of the remote sensing products as well as bias in inferred runoff and precipitation and challenges with water budget closure. I think this discussion as well as the motivation provided in the paragraph starting on Page 3 Line 25 should be expanded and should appear sooner in the introduction.

R7: The first paragraph demonstrates the importance and challenges existing in current water budget estimation and how this study was motivated and supported from different organizations. And then followed by the description of various data sources' strengths and weakness. The authors think this is a logic way to organize the introduction part but thanks for the reviewer's different opinion in structuring the paper.

C8. Section 2 should be expanded to provide a better summary of the strengths and weaknesses of the different datasets without relying so heavily on the supplemental material (e.g. page 5 line 14 and section 2.1.2 paragraph 1). I think it's fine to refer to the supplement for the details of these datasets but additional discussion is needed in the main text to explain to the reader the strengths and weaknesses of these approaches and why they were chosen. For example, it is important to clearly explain here the difference between satellite data, reanalysis products and

land surface models including what goes into each and what assumptions they rely on before comparisons are made. Some of this information comes up in the discussion of differences but it would be helpful to outline approaches upfront first.

R8: We thank the reviewer's suggestion. The challenge in presenting this work is (in part) the massive amounts of information that is fused for the CDR product. Finding the balance between material for the paper, and material for the supplement, is complicated. On the one hand, the paper can get dragged down if too many details are included. But on the other hand, having too high-level description may leave some readers (or yourself) feeling the description in the paper is incomplete. It's always a balance. As for the strengths and weaknesses of the various types of data products (satellite data, reanalysis products and land surface model output) for global products, a thorough handling is beyond the scope of this paper, but would be an excellent review paper along the lines of Wada et al. (2017) referred to above. For the work presented here, supplement I provides the basic information (e.g. resolution, brief algorithm) of each product and its reference, while section 2.1 analyzes and describes the seasonal cycle and difference existing between different data source at continental and basin scales. The authors think it is a reasonable way to organize the manuscript and would like to keep it as is.

C9. The figures could be improved to provide more quantitative metrics of performance especially with respect to spatial and temporal variability. For example, Figure 11 maps all of the water balance components globally in a single figure for multiple time periods but each subplot is so small it's very difficult to note the connections the authors are discussing. Some cutouts or regional assessments would be useful. Also, Figures 2-9 are repetitive and I think some of these could be moved to the supplemental material or different plotting approaches could be tested to summarize this information with less figures.

R9: Thanks for the reviewer's suggestion. Since this study is focused on global water budget closure, we believe that global maps are very necessary (e.g. Figure 11). In addition to the overview global maps, figures 2-9 further provide regional information at continental scale and basin scale for precipitation, *ET*, runoff and *TWSC*. Intuitively Figures 2-9 look repetitive but they actually provide different information, which is key to this study.

**Specific Comments:**
1. The list of satellite products page 2 line 25 would be easier to follow in table form.

The satellite products mentioned in the introduction at Page 2 are listed in the text in order to give the reader a general introduction of the available satellite products for each water budget variables. References were also provided in case the readers are interested in the details. This study does have a comprehensive table that includes all the data sets used in this study (Table 1). Including any data sets that exists but not used seems unnecessary.

2. Page 4 lines 3: I think before the paragraph laying out the advantages of this approach a more thorough explanation of the weaknesses of previous approaches would be helpful. For example, the first reason given here is the expanded use of the Constrained Kalman filter; however, the current limitations of the Kalman filter have not been explained.

Thanks and we have added it into the text.

"In this study, the Constrained Kalman Filter (CKF), which is a simplified version (non-ensemble) of the constrained ensemble Kalman filter (CEnKF, (Pan and Wood, 2006)), is chosen to close water balance. The CKF is a non-ensemble form, and is a standalone procedure after a

regular Kalman Filter update, thus it is ideal for closing water balance without filtering or data assimilation."

3. Table 1 should clearly differentiate land surface models from remote sensing products.
As mentioned in the text, the only land surface model applied in this study is VIC thus it should already be clear from Table 1.

4. Page 5 lines 2-7: This is very detailed for this intro to this section. I think it would be better to keep this high level, and provide an overview of the general approach and the organization of section 2 for the reader here.
The structure of section 2 is clear by looking at the sub-title. Those details (Page 5 lines 2-7) are necessarily pointed out for better understanding of the following section (e.g. why the plots are for different period?).

5. Figure 2: A more detailed caption explaining the acronyms and the difference between the grey line and the colored lines is needed. Some of this is included in the * points. You should rewrite these to incorporate all of this into a single caption. This is also true of the subsequent figures, which should be adjusted accordingly.
Thanks, the captions were changed accordingly.

6. For figures 2- 9: I think it would make more sense to plot the standard deviation rather than the coefficient of variation. The CV values clearly display a seasonal pattern caused by dividing by the mean. Since this information is already provided in the colored lines in my opinion it would be easier to understand if the grey line just showed standard deviation. This would also address the 'abnormal high spread' noted on page 5 line 25.
The authors prefer to use the CV, instead of the standard deviation, to quantify the variability of the data. A lower standard deviation does not infer less variable data (relative to the mean) due to different mean values in different months within a year. We believe that the CV can help to understand the relative variability of the data. The grey line is an uncertainty band in terms of percentage to quantify the normalized spread among data sets.

7. Section 2.1.1: Some aggregated statistics of differences in total precipitation for the major basin would be helpful to quantify the overall differences between approaches.
The authors agree that aggregates statistics like what we have in Table 3 and 4 could be developed. While total precipitation for major basin would be helpful, we're trying to focus the study globally. There is certainly potential for follow-on studies that can consider regional or major basins, but to do that here would make the paper exceedingly long.

8. Page 6 line 12: The derivation of the other four satellite products is described but not the GLEAM dataset.
GLEAM was described ahead of the other four satellite products and its reference is provided as well.

9. Section 2.1.3: I think this section should include a description of how runoff is calculated in each model and the strengths and weaknesses of each approach and their systematic biases.

We do not think it is necessary to go into that detail as the rainfall-runoff procedures in different models can be found in their corresponding references for interested readers. But we do compare and describe the runoff simulation performances among those three land surface models themselves as well as against GRDC ground observations in the manuscript.

10. Page 7 line 6: Can you be more specific about what type of discrepancy you are referring to (i.e. a low bias)?

11. Page 7 line 7: Can you be more specific about the type of 'disagreement' you are referring to? For 10 and 11, the authors have modified the text:

"Noah shows opposite seasonal cycle against VIC and CLM in Europe and North America, which include high latitude regions (Figure 6). Unlike VIC and Noah, CLM almost shows no seasonal cycle in Oceania (Figure 6)."

12. Figure 13 should be figure 8 since it gets referred to after Figure 7
This was corrected in the manuscript, thanks.

13. Page 7 line 14: Should be 'capture'
This was revised in text, thanks.

14. Page 7 line 14-15: This is unclear, can you expand on the uncertainty estimates you are referring to here?
Here uncertainty refers to spread or say, standard deviation among different data sources. This has been added into the text.

15. Page 7 line 19: It would be helpful to define 'total water storage change' and 'total water storage anomaly' explicitly here before getting into this discussion.
$TWSC$ measures the changes in total water storage during a specific period unit. TWSA is defined in the manuscript in Page 7 (and repeated here) while $TWSC$ is clearly defined by the equations (2) & (3).
"The GRACE monthly total water storage anomaly ($TWSA$) time series, which are anomalies relative to the 2004-2009 time-mean baseline from ReLease 05 (RL05) that are processed by three centers, Geoforschungs Zentrum Potsdam (GFZ), Center for Space Research at University of Texas, Austin (CSR), and Jet Propulsion Laboratory (JPL), …"

16. Page 7: Equation 2 is not necessary in my opinion since this approach wasn't used.
The reviewer is correct, equation (2) was not applied, but we would like to keep it in the text in order to give the readers a clear idea of the common approaches in calculating $TWSC$.

17. Page 7 line 20: It would be helpful to explain what the significant differences in these three processing centers are.
Different parameters and solution strategies were explored and applied by these three processing centers and the differences between the centers were very small and have generally decreased over the Releases (https://grace.jpl.nasa.gov/data/choosing-a-solution/). For detailed differences between different centers, please refer to (Sakumura et al., 2014), in which the authors found that the ensemble mean (simple arithmetic mean of JPL, CSR, GFZ) was the most effective method in reducing the noise in the gravity field solutions within the available scatter of the solutions.

18. Page 8 Line 10: It sounds like you are using the ensemble mean of GRACE here for future TWSC analysis and not using VIC at all but I don't think this is the case.
To make it clear, we have modified the sentence as:
"Therefore, the ensemble mean of the $TWSC$ from GFZ, CSR, and JPL is taken as the best $TWSC$ product derived from GRACE, and this is used later in the water budget analysis together with $TWSC$ from VIC."

19. Page 9 lines 3-10: Some demonstration of the impact of this adjustment on the time series would be helpful here given that the authors argue it is a 'key step' for temporal consistency.
This is done in order to avoid the "jump" between different sub-periods and guarantee the temporal consistency in the merged data time series. We have also added one sentence into the text to further demonstrate the impact of this adjustment.
"This "data consistency" approach aims to avoid the "jump" in the merged precipitation time series in the year 1998 when the CSU became available. The same procedure is then applied to adjust the data consistency for $ET$ during 2008-2010 and $TWSC$ during 1984-2002. We contend that this is a key step, as the temporal consistency of the CDR will impact the reproduction of historical hydrological extremes and the analysis of long-term trends for all the available water budget variables. "

20. Page 10 lines 22-23: Globally mean TWSC may be small but this does not mean local changes are small and if the point is 0.5degree resolution I think this could be a limitation. Some discussion of spatial variability would be helpful here.
Yes, that's true. We have extended this a little bit into the text.
"Given the good agreement in $TWSC$ between VIC and GRACE (Figures 9 and 10), the impact of such a subjective error assignment is relatively small. But for high-latitude basin such as Yukon where VIC and GRACE have relatively large discrepancy, the error is relatively high."

21. Page 13 Lines 6-7: What does it mean to be 'filtering out those basins with nonsignificant correlations'? This sounds like an additional step beyond the filtering for different anthropogenic impacts. What was the threshold for this filtering and how many points were filtered because of it?
33 medium basins (out of 362) and 36 small basins (out of 862) were filter out by running a test of significance to remove those catchments with non-significant correlations between GRDC runoff observations and CDR runoff records in order to remove those basins such as Indus and Senegal which might have incorrect observational data. This incorrect observational data records were also discussed in the text:
"Note that the seasonal peaks from Noah and VIC are in agreement for the Indus basin but their peaks precede the peak from the GRDC observations, which strangely happen in November. Comparing to other studies for the Indus River (Bookhagen and Burbank, 2010) show that the discharge peak occurs in the summer time , which is consistent with VIC and Noah. Likewise for Senegal River, records from regional studies (Andersen et al., 2001) and (Stisen et al., 2008) show runoff peaks in August to September instead of April to May from the GRDC record."

22. Page 14 lines 9-10: Even though ET is most dominant during the summer I think that the verification should not be limited to the warm season without further justification.

The authors needed to do a lot of data 'cleaning' before applying the observational *ET* from the flux tower for validation. For example, there are a lot of missing data in the raw flux net data we received, particularly during winter. In order to select reasonable flux tower observations for effective validation at a monthly scale (which is the temporal resolution for the CDR in this study), those months with less than 70% of their data records were removed. After a careful check, the validation was only done during summer season. And this is further explained in the text:

"The raw data are at 3 hourly and the most complete data were recorded during the warm seasons. Therefore, the comparisons are made only over the summer (warm) seasons by filtering out those years with less than 70% data based on the data availability at each tower."

---

## Author Response (AR2)

This is the second review of 'A Climate Data Record (CDR) for the global terrestrial water budget: 1984-2010'. I thank the authors for their response to reviewer questions and comments and for their revisions to the manuscript. A few minor points still remain to be clarified, however. These are detailed below.

C1. Section 2.2.1, page 8-9: I thank the authors for their explanation in their response to the question about the use of "optimal" to describe the merging technique. Based on this, it seems that it is perhaps more accurate to qualify this "optimality" as constrained by or conditional on limited data availability rather than being the optimal approach. A few minor edits, consistent with the authors' response to R1-Comment 2, in the abstract and section 2.2.1 would clarify this point.

R1: Thanks for the reviewer's suggestion, we've edited it accordingly in the abstract at Page 1, line 18 -23, and section 2.2.1 at Page 8, line 30-33 and Page 9, line 1-3.

"Conditioned on the current limited data availability, a systematic method is developed to optimally combine multiple available data sources for precipitation ($P$), evapotranspiration ($ET$), runoff ($R$) and the total water storage change ($TWSC$) at 0.5° spatial resolution globally and to obtain water budget closure (i.e. to enforce $P - ET - R - TWSC = 0$) through a Constrained Kalman Filter (CKF) data assimilation technique under the assumption that the deviation from the ensemble mean of all data sources for the same budget variable is used as a proxy of the uncertainty in individual water budget variables."

"There is no best estimate or observation of each individual water budget component at the grid scale over the globe due to the limited spatial coverage of in-situ measurements. This is especially true for evapotranspiration observations from the flux tower networks. Thus, the limited availability of gridded ground observations makes it impossible to quantify the error in each water budget component. Therefore, in this study, the deviation from the ensemble mean of all data sources for the same budget variable is used as a proxy of the uncertainty/error in individual products. The merging procedure for each budget component is a weighted, averaging where the optimal merging weight $w_i$ is given by the following equation…"

C2. Page 9, Lines 31-32: It would be helpful here to state that the assumed 10% error in VIC runoff is based on the authors' experience and judgment "given there is no global grid level (0.5 degree in this study) runoff observations to quantify the error."

R2: Thanks, and this is further clarified at Page 10 line 2-3.

"And this is highly empirical based on the authors' knowledge and confidence about the VIC model calibration given there is no global grid level (0.5 degree in this study) runoff observations to quantify the error."

C3. Page 11, lines 25-30: I recognize the difficulty in accounting for groundwater extraction and management globally. As the authors note, there are regions (California Central Valley, US High Plains, Iran, etc) where historic regional storage declines challenge the assumption of zero long

term TWSC. For completeness, a explanation of the consequences of this assumption on the CDR results for these areas, either in section 3.2 or in the discussion in section 5, seems warranted.

R3: Thanks and we have added explanations of the potential consequences by neglecting the groundwater extractions at page 12 line 3-4.

"The "zero TWSC" assumption would potentially introduce local/regional bias into the water budget estimates in the regions with groundwater depletions."

C4. Page 13, line 22: In response to R1-comment 12, the authors state that the filtering out of basins with non-significant correlations "was done in order to remove those basins such as Indus and Senegal which might have incorrect observational data." This was not immediately apparent from the explanation on page 13 of the revised manuscript – please edit to include this point for clarity.

R4: Thanks for the reviewer's comment. We've edited this point at page 13 line 29-32.

"A test of significance test was conducted to remove those medium and small basins with non-significant correlations between GRDC runoff observations and CDR runoff records.  This was done in order to remove those basins such as Indus and Senegal which might have incorrect observational data."

C5. Page 16, lines 8-12 and Figure S7: I thank the authors for the additional text and plots addressing the question of inter-annual variability in the CDR. However, this section is a little unclear. Please define SPI and provide more detail in the Figure S7 caption to indicate what is shown in the plots (i.e which parts are from the CDR?)

R5: Thanks. We have edited the caption of Figure S6 in order to make this clear.

"Figure S1 1998-1999 US drought captured by CDR in terms of 6-month SPI and drought extends calculated from CDR precipitation"

And the SPI is also defined in the text at Page 16 line 17 -18.

"Figure S7 further provides an example of how the CDR captured the 1998-1999 US drought in terms of Standardized Precipitation Index (SPI) and drought extends calculated from CDR precipitation."

C6. Figures – general: The number of figures in the main manuscript makes the main point of this manuscript less clear. Some of the figures could be moved to the supplemental information to better emphasize the results of the study. For example, the data product comparisons (Figures 2-10) could be limited to just continental (or river basin) plots and the remainder moved to the SI.

R6: Thanks for the reviewer's suggestion. We understand the reviewer's concern and we have also struggled with the display of the figures. After careful consideration, we think it would be better to keep the figures as they are as we have a large portion of text describing and discussing the

seasonal cycles at both continental and basin levels. Therefore, figures 2-10 would be better to be all kept in the main manuscript instead of the supplement.

C7. Figure 2: The caption references TMPART but this seems inconsistent. Should it instead reference the CSU dataset?

R7: Thanks and we have changed TMPART into CSU in the caption of Figure 2.

C8. Figure 11: Please provide a more descriptive caption to accompany this flowchart.

R8: Thanks and we have changed the caption of Figure 11 into "Flowchart describes the progress of data pre-processing, error analysis, water balance constraint and multi-scale water budget analysis"